# ATR regulates neuronal activity by modulating presynaptic firing

Murat Kirtay[1], Josefine Sell[2], Christian Marx[1], Holger Haselmann [2], Mihai Ceanga[2], Zhong-Wei Zhou[1,3], Vahid Rahmati[2], Joanna Kirkpatrick[1], Katrin Buder[1], Paulius Grigaravicius[1], Alessandro Ori [1], Christian Geis[2✉] & Zhao-Qi Wang [1,4✉]

Ataxia Telangiectasia and Rad3-related (ATR) protein, as a key DNA damage response (DDR) regulator, plays an essential function in response to replication stress and controls cell viability. Hypomorphic mutations of ATR cause the human ATR-Seckel syndrome, characterized by microcephaly and intellectual disability, which however suggests a yet unknown role for ATR in non-dividing cells. Here we show that ATR deletion in postmitotic neurons does not compromise brain development and formation; rather it enhances intrinsic neuronal activity resulting in aberrant firing and an increased epileptiform activity, which increases the susceptibility of ataxia and epilepsy in mice. ATR deleted neurons exhibit hyper-excitability, associated with changes in action potential conformation and presynaptic vesicle accumulation, independent of DDR signaling. Mechanistically, ATR interacts with synaptotagmin 2 (SYT2) and, without ATR, SYT2 is highly upregulated and aberrantly translocated to excitatory neurons in the hippocampus, thereby conferring a hyper-excitability. This study identifies a physiological function of ATR, beyond its DDR role, in regulating neuronal activity.

[1] Leibniz Institute on Aging - Fritz Lipmann Institute (FLI), Jena, Germany. [2] Section of Translational Neuroimmunology, Department of Neurology, Jena University Hospital, Jena, Germany. [3] School of Medicine (Shenzhen), Sun Yat-Sen University, Guangzhou, China. [4] Faculty of Biological Sciences, Friedrich Schiller University of Jena, Jena, Germany. ✉email: Christian.Geis@med.uni-jena.de; Zhao-Qi.Wang@leibniz-fli.de

The genome is constantly attacked by exogenous and endogenous agents, including DNA replication fork stall, DNA strand breaks, metabolite, reactive oxygen species (ROS), environmental toxins and radiation. DNA damage is one of the most critical threats to the organisms. Thus, the cell has evolved a delicate, yet robust DNA damage response (DDR) mechanism which controls DNA repair, cell-cycle progression (checkpoint), apoptosis and transcription, to protect the genome integrity[1–3]. Ataxia Telangiectasia and Rad3-related (ATR) protein belongs to the family of phosphoinositide three-kinase-related protein kinases (PIKKs), which includes other DDR molecules like ATM and DNA-PK. ATR is primarily activated by replication stress and DNA single stranded breaks (SSBs). Activated ATR can phosphorylate many substrates, but mainly its classical substrates TopBP1 and Chk1, and also initiates ATR-Chk1-mediated S-phase checkpoint[4,5].

ATR and its substrates CHK1 and TopBP1 are all essential to cell survival and constitutive deletion of these proteins causes lethality in animal models, as they are required to handle stalled replication fork damage in the S-phase checkpoint by modulating a serial of DDR cascade, including damage signaling and DNA repair[6–10]. An inducible deletion of ATR in postnatal tissues, which spares animal life, affects many somatic tissues harboring proliferating cells[11]. Moreover, a specific knockout of ATR in the central nervous system (CNS) of mice shows a crucial function of ATR in neuroprogenitor proliferation and DDR-mediated cell death[12,13]. These biological functions of ATR derived from cellular and animal models highlight the importance of ATR-CHK1-mediated DDR in highly replicative cells. Hypomorphic mutations of the *ATR* gene are responsible for human ATR-Seckel Syndrome (ATR-SS), a chromosome instability disorder characterized by dwarfism, severe microcephaly, growth retardation and intellectual disability[14–18]. Expression of humanized hypomorphic allele of *ATR* in mice ($ATR^{S/S}$) phenocopied many characteristics of ATR-SS, including smaller body size and microcephaly[19]. Due to the essentiality of the *Atr* gene in vivo, the role of ATR in the pathogenesis of postnatal neurological and cognitive defects remains unknown.

The neurological symptoms, such as microcephaly, learning deficits and intellectual disabilities of ATR-SS patients and animal models, may well reflect abnormal neuron activities[15–17,20] and thus suggest a potential role for ATR in postmitotic neurons. In general, synaptic function and homeostasis is crucial for neuronal network activity. Ligand-gated cation channels and anionic channels are crucial to regulating membrane excitability[21–23]. Large scale human interactome studies (The BioPlex Network) showed that ATR potentially interacts with two sodium channel subunits, namely β2 (SCN2B) and β3 (SCN3B)[24,25] and is also reported to associate with the presynaptic proteins VAMP2 and synapsin-1 in cultured neurons[26]. However, the biological significance of these interactions in neuronal activities and the pathogenesis of ATR-SS patients remains elusive.

To help decipher the physiological function of ATR in postmitotic tissues and in the disease course of ATR-SS, we developed mouse models with specific ATR deletion in inhibitory and excitatory postmitotic neurons, respectively. We found, surprisingly, that ATR loss does not impinge on brain formation and architecture, but alters intrinsic activity of both types of neurons. Further, ATR deletion in forebrain excitatory neurons compromises presynaptic functionality and neurotransmitter release, thereby elevateing neuronal excitability and leading to increased epileptiform activity. ATR interacts with presynaptic vesicle partners SYT2 and PROT and regulates their expression in excitatory neurons. These defects are apparently independent from ATR-mediated DDR. Thus, we discover a physiological function of ATR in neuronal excitability and presynaptic function.

## Results

### ATR deletion is compatible with cerebellar development but causes locomotor dysfunction.
Deletion of ATR in neuroprogenitors results in early postnatal lethality, around day 7 after birth[12,13]. To investigate the function of ATR in a specific population of postmitotic neurons, we first generated a conditional knockout mouse model, wherein ATR was deleted in Purkinje cells (PCs) of the cerebellum (ATR-PCΔ) by crossing $ATR^{flox}$ mice[11] with L7/pcp2-Cre transgenic mice[27]. ATR-PCΔ mice were born normally and showed no obvious phenotype in the period of 2 years. The gross morphology of ATR-PCΔ brains as well as the lobular and anterior-posterior organization of the cerebellum were normal (Fig. 1a and Supplementary Fig. 1a), despite a great reduction of the ATR protein in cerebella (Supplementary Fig. 1b). Immunostaining of PCs and Bergmann glia by Calbindin and GFAP antibodies, respectively, detected no obvious difference of the cerebellar morphology between mutants and controls (Fig. 1b). The total PC number and the thickness of molecular layer in the cerebellum were very similar between ATR-PCΔ brains and controls at young (3–4-month-old), mid (6-month-old) and old (18–20-month-old) age (Fig. 1c, d and Supplementary Fig. 1c).

Despite apparently normal structure and morphology of ATR-PCΔ cerebella, these mutant mice showed a striking locomotor dysfunction. They spent significantly less time on the accelerated rotarod compared to controls (Fig. 1e), which was especially obvious in old mice (Fig. 1f). Moreover, in multiple trials of five consecutive days, these mice failed to improve their rotarod performance, in contrast to the control group. The tests indicate a defect in grab strength or motor coordination, and learning, with the declined locomotor performance further confirmed by the crossing beam test. The ATR-PCΔ cohort needed more time to cross the beam ($p = 0.072$) (Fig. 1g and Supplementary Videos 1 and 2), with the number of hindlimb slips and missteps during the walk higher than for the control mice (Fig. 1h)—an indication of miscoordination of the limbs and defective body balance. As such, ATR deletion in PCs leads to deficiency in locomotor coordination without affecting the architecture and morphology of the cerebellum.

### ATR maintains the intrinsic activity of Purkinje cells.
The PCs are intrinsically active inhibitory neurons that conduct the main output of the cerebellar neuronal circuit to control locomotor function[28]. The normal structure and morphology of the cerebellum in ATR-PCΔ mice prompted us to investigate whether ATR deletion causes any defects of PCs' neuronal activity. To this end, we performed loose-patch electrophysiological recordings on PCs of old mice to evaluate their intrinsic activity (Fig. 2a). We found ATR-deleted PCs to have a greatly increased spontaneous spiking frequency compared to controls ($p < 0.001$, Fig. 2b, c), with smaller interspike intervals (ISI) ($p = 0.006$, Fig. 2d). Yet, firing regularity was unaffected as measured by the coefficient of variation (CV) of ISI (Fig. 2e)—findings that indicate a pathologically enhanced intrinsic PC neuronal activity without ATR. We next monitored the parallel fiber-evoked (PF) activity in PCs and measured the PC firing latency following the first evoked spike after PF stimulation. ATR-deleted PCs responded to PF stimulation similarly to controls, judged by normal spiking delay and variance (Fig. 2f, g). Consistently, VGLUT1 immunostaining revealed a normal structure and intensity of the PFs in the molecular layer of the ATR-PCΔ cerebellum (Supplementary Fig. 1d). Taking these observations together with locomotor dysfunctions, ATR regulates the intrinsic activity of PCs to coordinate the locomotor function, via deep cerebellar nuclei and brainstem vestibular nuclei[29,30].

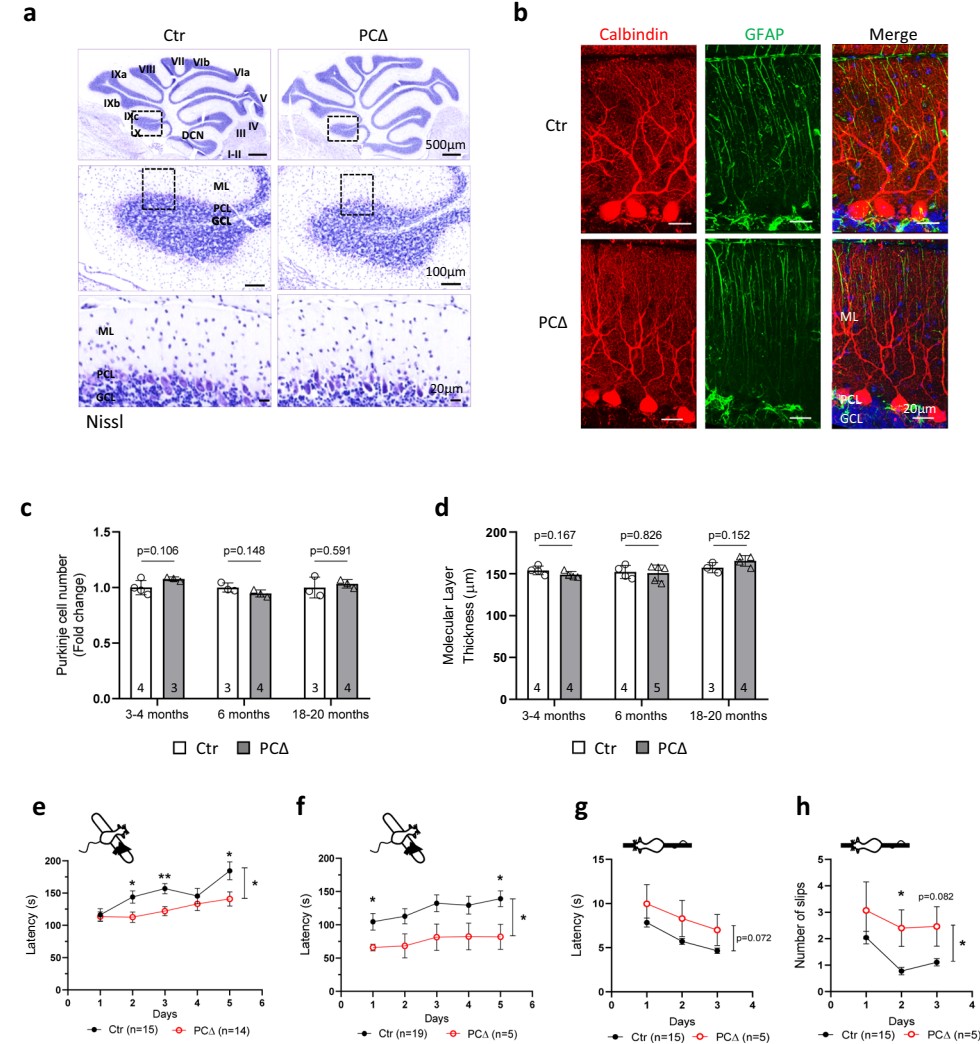

**Fig. 1 Deletion of ATR in Purkinje cells leads to locomotor dysfunction and learning defects in mice. a** Sagittal brain sections of the control (Ctr) and ATR-PCΔ (PCΔ) brains were stained with cresyl violet (Nissl staining). Upper panel shows the complete cerebellum. The magnified view of the frames in the upper and mid panels are shown in the lower panels, respectively. The images are representative from 3 to 4 mice of each genotype analyzed. DCN Deep Cerebellar Nuclei, ML Molecular Layer, PCL Purkinje Cell Layer, GCL Granular Cell Layer. **b** Sagittal sections of 3-month-old control and ATR-PCΔ mice were stained with DAPI, Calbindin and GFAP antibodies to label nuclei, Purkinje cells and Bergmann glia, respectively. ML Molecular Layer, PCL Purkinje Cell Layer, GCL Granular Cell Layer. **c** Quantification of total Purkinje cell number compared to control at the indicated age. The data is represented as fold changes compared to the control group. The number of mice is indicated within the bar. Error bars indicate SD. Student's *t*-test (unpaired, two-tailed) is performed for the statistical analysis. The *p* values are indicated in the graphs. **d** Quantification of the thickness of the molecular layer of the cerebellum of mice at the indicated age. The number of mice is indicated within the bar. Student's *t*-test (unpaired, two-tailed) is performed for the statistical analysis. The *p* values are indicated in the graphs. **e** The rotarod performance of 4–9-month-old mice on five consecutive days. Error bars indicate SEM. *P* values: day 2, *p* = 0.014; day 3, *p* = 0.002; day 5, *p* = 0.028; group comparison *p* = 0.034. **f** The rotarod performance of 18–20-month-old mice on five consecutive days. Error bars indicate SEM. *P* values: day 1 *p* = 0.025; day 5, *p* = 0.024, group comparison *p* = 0.041. **g** The beam walking performance was tested in the 18–20-month-old group. The latencies on 1 cm width beam platform on three consecutive days are shown. Error bars indicate SEM. *p* = 0.072. **h** The number of hindlimb slips during the beam walking performance test. Error bars indicate SEM. *P* values: day 2, *p* = 0.015; group comparison, *p* = 0.017. Two-way repeated measures ANOVA with Sidak's multiple comparisons test or MWU within individual days was performed for the statistical analysis in **e**–**h**. *\*p* < 0.05, *\*\*p* < 0.01. The number of mice (*n*) tested are indicated within the graph legend. Source data are provided as a Source Data file.

**Deletion of ATR in glutamatergic neurons results in epileptic seizures.** Next we investigated whether ATR has a role in regulation of neuronal activity of excitatory neurons. ATR was deleted in excitatory glutamatergic neurons of mouse forebrains (ATR-FBΔ) by crossing *ATR*^flox mice with CamKII-Cre mice[31]. The gross morphology and structure of ATR-FBΔ brains were normal in young (3-month-old) (Fig. 3a, b) and 10-month-old aged (Supplementary Fig. 2a). Western blot analysis confirmed the efficient deletion of ATR in different regions of the forebrain

(Supplementary Fig. 2b). Although the brain weight of ATR-FBΔ animals was slightly reduced (Supplementary Fig. 2c, d), the body weight and the ratio of brain/body weight showed no significant differences between ATR-FBΔ and control animals (Supplementary Fig. 2e). Furthermore, we did not detect any apoptotic signal in the hippocampus of ATR-FBΔ mice (Supplementary Fig. 2f).

Notably, ATR-FBΔ mice began to exhibit sporadic non-lethal epileptic seizures at around 10 months old, which may have been

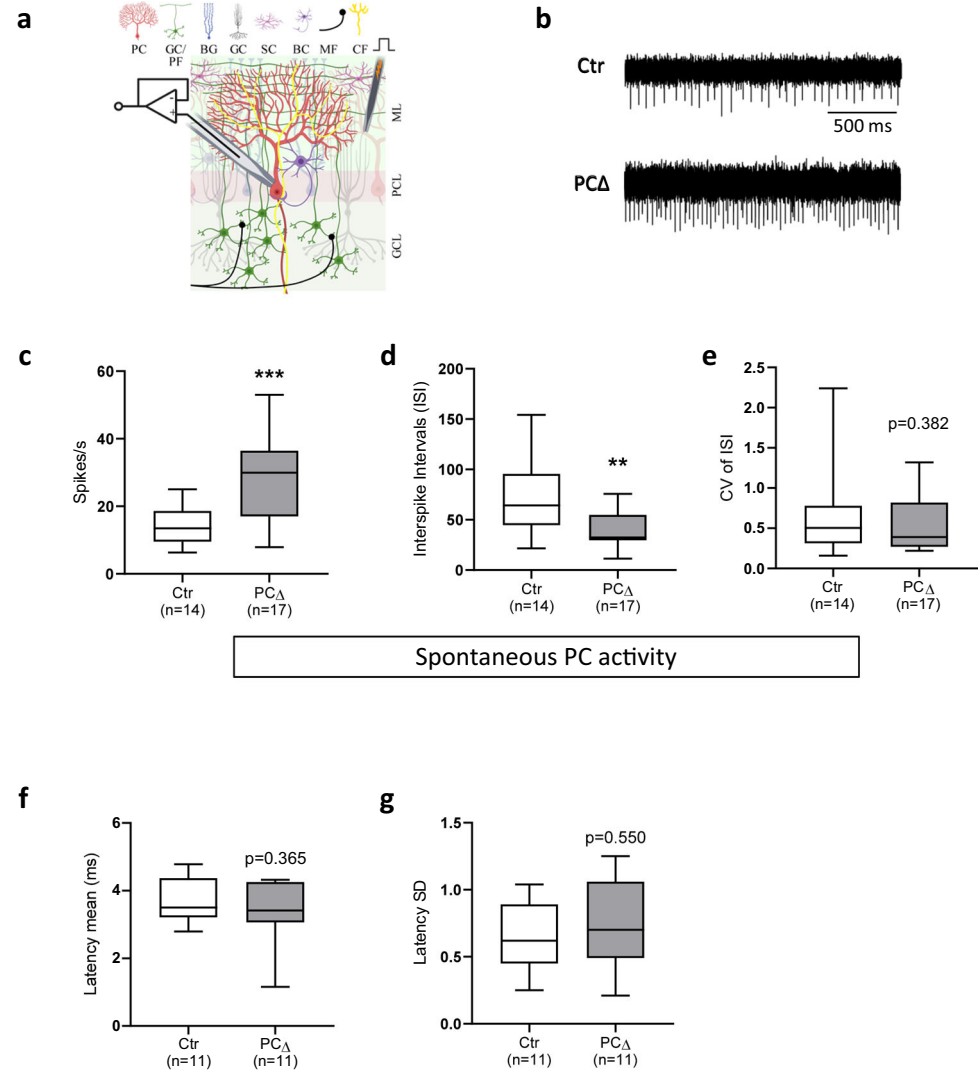

**Fig. 2 ATR-PCΔ neurons have abnormal intrinsic activity. a** Schematic representation of cerebellar loose-patch electrophysiology measurements on Purkinje cell soma and stimulation of parallel fiber-Purkinje cells synapses. PC Purkinje cell, GC/PF granule cell with parallel fibers, BG Bergmann glia, GC Golgi cell, SC stellate cell, BC basket cell, MF mossy fiber, CF climbing fiber. **b** Sample traces from the passive activity recordings of Purkinje cells of controls (Ctr) and ATR-PCΔ (PCΔ) mice. **c** Passive activity as shown by spikes/s revealed increased spiking frequency in ATR-PCΔ neurons. $p < 0.001$. **d** Interspike intervals (ISI in ms). The ISI includes only spikes with intervals <2 s. $p = 0.006$. **e** The coefficient of variance (CV) of ISI. CV of ISI is an indicator for the reliability of the cells' tonic firing. $p = 0.382$. **f** The firing latency measured from the evoked spike to next spike (in ms). $p = 0.365$. **g** The standard deviation (SD) of this latency between all sweeps of one recorded cell. $p = 0.550$. MWU test (unpaired, two-tailed) is performed for the statistical analysis. **p < 0.01, ***p < 0.001. The number of neurons measured is indicated within the graph. Box plots in (**c**–**g**) indicate the median (middle line), whiskers as max value (top) and min value (bottom) and hinges as 25 percentile (top) and 75 percentile (bottom). Source data are provided as a Source Data file.

triggered by external stimuli such as handling and cage changing. Epileptic mice showed generalized tonic-clonic seizures, lasting for 10–30 s and terminating spontaneously followed by a hypoactive post-ictal phase (Supplementary Video 3, 4, 5). GFAP antibody staining of brain sections of epileptic ATR-FBΔ mice revealed an increase of immunoreactivity mainly around the dentate gyrus area (Fig. 3c) and also in the cortex (Supplementary Fig. 3a), indicating astrogliosis. Besides, NeuN antibody staining detected less granule cells in the dentate gyrus and also granule cell dispersion (Fig. 3c). Immunostaining of hippocampal mossy fibers with antibodies against zinc transporter 3 (Znt3) and neurofilaments with SMI312 confirmed astrogliosis, namely mossy fiber sprouting[32], in the inner molecular layer and granular layer of the dentate gyrus (Supplementary Fig. 3b). This

was accompanied by infrapyramidal mossy fiber bundles extending through the CA3 region in ATR-FBΔ hippocampi (Fig. 3d). Again, these abnormalities were absent in non-epileptic mutant mice (Supplementary Fig. 3c). These seizure-related phenotypic changes, e.g. neuron loss in the dentate gyrus or the CA1 region and granule cell dispersion, are connected with mossy fiber sprouting and reactive gliosis and glial scars[33]. As these changes are present only in epileptic ATR-FBΔ mice, reactive gliosis and neuronal loss are most likely secondary to seizures and not mediated by neuronal ATR deletion per se.

**ATR deletion increases seizure susceptibility in the hippocampus.** Epileptiform activity often initiates in the CA3 region of

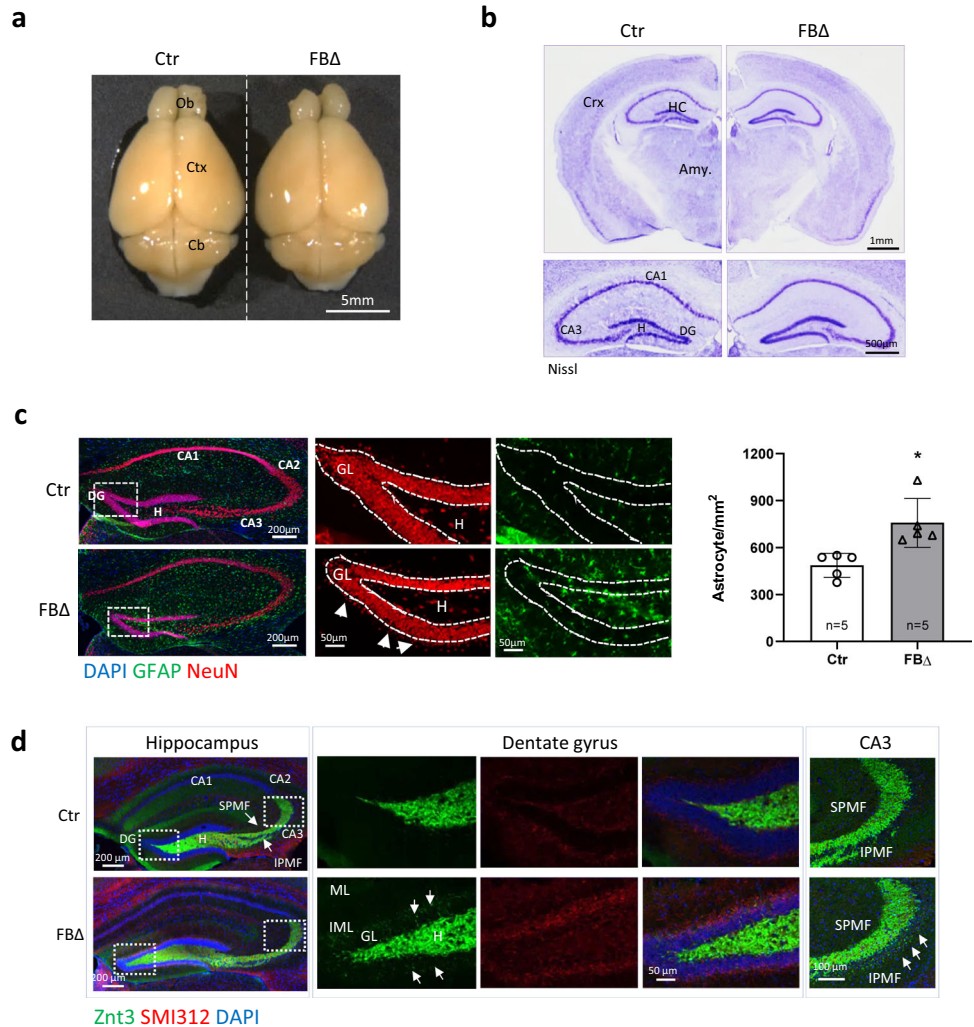

**Fig. 3 ATR deletion in excitatory neurons of the mouse forebrain (ATR-FBΔ) leads to developments of epileptic signs. a** Dorsal views of the brains from 3-month-old control (Ctr) and ATR-FBΔ (FBΔ) mice. Ob Olfactory Bulb, Ctx Cortex, Cb Cerebellum. **b** Coronal brain sections of 3-month-old control and ATR-FBΔ brains were stained with Nissl. The upper panel shows the complete half hemisphere and the lower panel displays the magnified images of the hippocampal regions. The images are representative from 3 to 4 mice per genotype analyzed. Ctx Cortex, HC Hippocampus, Amy. Amygdala, DG Dentate Gyrus, CA Cornu Ammonis, H Hilus. **c** Hippocampal astrogliosis in ATR-FBΔ mice. Coronal sections from control and ATR-FBΔ littermates at 10–12 months old after epileptic seizures were stained with DAPI, GFAP and NeuN to label nuclei, astrocytes and post-mitotic neurons, respectively. Right panels show magnifications of the dentate gyrus. Neurons are leaving the dentate gyrus (arrows) and expression of GFAP is markedly increased. CA Cornu Ammonis, DG Dentate Gyrus, H Hilus, GL Granular Layer. The graph shows the quantification of GFAP-positive cells within the hippocampal area from at least 2–3 sections per animal. The number of mice used is indicated within the bar. Data are presented as mean values of ± SD. Student's *t*-test (unpaired, two-tailed). $p = 0.013$. *$p < 0.05$. **d** Mossy fiber sprouting in the hippocampus of 10-month-old epileptic ATR-FBΔ mice. Coronal sections of brains were stained with DAPI, ZnT-3 and SMI312 antibodies to label nuclei, mossy fibers and axonal neurofilaments, respectively. White arrows indicate the mossy fiber sprouting in IML. Neurofilaments follow the same pattern of ZnT-3 staining. The magnified images of the white rectangles on the DG and CA3 areas are shown in the right panels. The images are representative from 3 mice per genotype analyzed. CA Cornu Ammonis, DG Dentate Gyrus, SPMF Suprapyramidal Mossy Fibers, IPMF Infrapyramidal Mossy Fibers, H Hilus, GL Granular Layer, ML Molecular Layer, IML Inner Molecular Layer. Source data are provided as a Source Data file.

hippocampus and spreads into other areas like CA1 or dentate gyrus[34,35]. To investigate ictogenesis at its site of origin, we recorded extracellular field potentials from the CA3c-d region in hippocampal slices of 3-month-old animals ($n = 16$ slices from 4 control mice, $n = 18$ slices from 4 ATR-FBΔ mice). We used $Mg^{2+}$ free solution containing 4 mM extracellular $K^+$ to induce preictal epileptiform discharges (PEDs), which precede ictal-like activity[35]. PED events were filtered to obtain information on their population field activity (PFA), fast ripple activity (FRA) and multiple unit activity (MUA) signals (Fig. 4a). Remarkably, ATR-deletion increased the level of spontaneous population activity in the CA3 (Fig. 4a–d). This was quantified as an overall increase in

the instantaneous PFA power throughout the recording, with most of the contribution coming from a significant increase in the power of the PEDs (Fig. 4c, d). Furthermore, the number of slices presenting high-activity PEDs was significantly higher in ATR-FBΔ mice compared to controls ($p = 0.012$, Fig. 4b) whereas the PEDs rate was similar for both groups (control: $0.43 \pm 0.05$ per sec, ATR-FBΔ: $0.42 \pm 0.05$ per sec; $p = 0.8231$).

To investigate the power of PEDs at different frequency components, we further analyzed the power spectral density (PSD) of these events. PSD of PEDs increased significantly in both PFA ($p = 0.026$, Fig. 4e, f) and FRA signals ($p = 0.023$, Fig. 4g), relative to that of their corresponding baseline activity

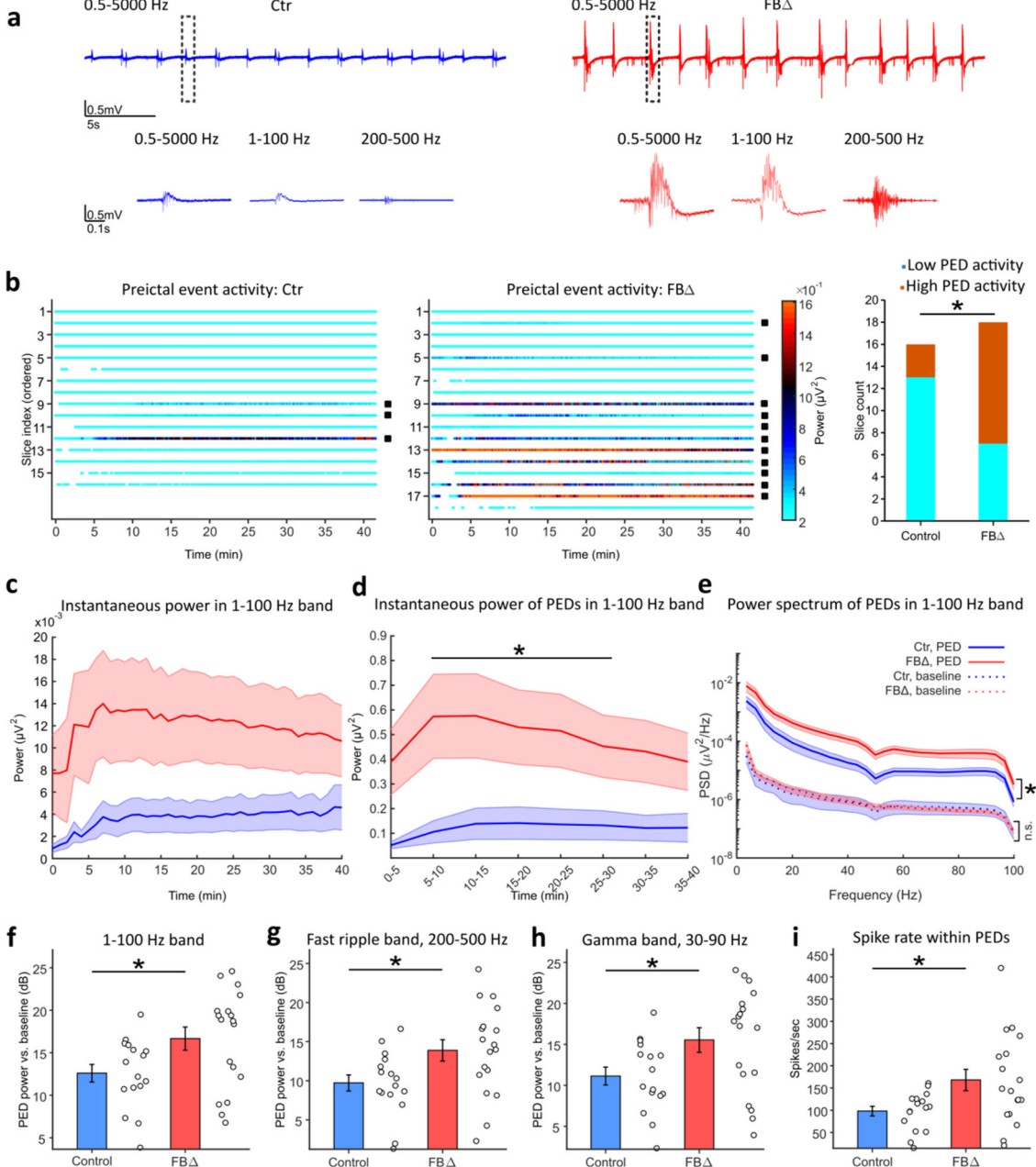

**Fig. 4 The CA3 region is susceptible to epileptiform activity in the ATR-deleted hippocampus. a** Deletion of ATR increases the amplitude of the preictal epileptiform discharges (PEDs). Example 30-second sections of 40-minute extracellular field potential recordings of PEDs in CA3. Zoom-in representative PEDs (0.5–5000 Hz) in population field activity band (PFA, 1–100 Hz) and fast ripple activity band (FRA, 200–500 Hz). **b** PEDs show higher instantaneous power in ATR-FBΔ (FBΔ) CA3 slices compared to controls (Ctr). Color-coded raster plots of the PED timing in all recorded slices. Colors represent the maximum instantaneous power of each PED in 1–100 Hz band. The slices are arranged in descending order, by the first slice having the highest number of detected PEDs. The slices containing PEDs with a power-level beyond 0.4 μV$^2$ (high-activity PEDs) are designated with a black square next to them. The ratio of slices with high-activity versus those with low-activity PED instantaneous power is increased after ATR deletion (right panel); Chi-square test (one-tailed), $p = 0.012$. **c** The instantaneous power of the whole field potential signal in 1–100 Hz band, computed within non-overlapping 1-min bins. The mean (solid line) ± SEM (shaded area). **d** Same as (**c**), but only for isolated PEDs (peak values) within 5-min bins. A one-tailed permutation test method of Cohen is used to compare the results of the two groups at each bin, with $p < 0.05$ (see Methods). **e** ATR-deletion leads to an increase in the power spectrum of PEDs over 1–100 Hz band. Power spectral density (PSD) of the PED events (solid lines), together with that of the baseline (i.e. non-PED epochs; dotted lines). **f** The summed baseline-normalized power presents an increase in the power of PEDs over 1–100 Hz shown in (**e**). Data presented as mean ± SEM. **g** ATR-deletion induces stronger fast ripples within PEDs. Same as (**f**), but for the fast ripple band (200–500 Hz). **h** Same as (**f**), but for the gamma band (30–90 Hz). **i** ATR-deletion increases the rate of spikes within PEDs, as detected in the multiple unit activity signal (MUA, > 500 Hz). Data are obtained from 16 slices of 4 control and 18 slices of 4 ATR-FBΔ mice and presented as mean ± SEM. Two-tailed t-test for (**e-i**): $p = 0.026$ for (**e**) and (**f**), $p = 0.023$ for (**g**), $p = 0.027$ for (**h**), and $p = 0.016$ for (**i**). *$p < 0.05$. Source data are provided as a Source Data file.

(i.e. non-PED epochs). Consistent with the notion that epileptic patients and rodents have higher power in the gamma band of their brain activity[36,37], we also found that the PEDs presented stronger gamma rhythms in 30–90 Hz band ($p = 0.027$, Fig. 4h). Furthermore, our analysis of the detected spikes within PEDs in the MUA signal revealed a significant increase in the rate of these spikes in ATR-FBΔ group ($p = 0.016$, Fig. 4i), which corroborates previous studies showing such an increase in MUA in epilepsy models[35,38]. Together, these findings indicate an increased susceptibility to epileptic activity in the hippocampal CA3 region already at young age of ATR-FBΔ mice, eventually resulting in spontaneous seizures.

**ATR regulates intrinsic neuronal activity.** Epilepsy is associated with aberrant synaptic transmission and neuronal activity[39,40]. The susceptibility of ATR-FBΔ mice to epilepsy strongly suggests that altered neuronal activity in mutant mice, which progresses during aging, renders neurons vulnerable to stress or environmental stimuli. We next investigated how ATR loss affects synaptic transmission and neuronal activity underlying epileptic seizures using young (3-month-old) animals prior to epileptic attacks. Whole-cell patch clamp recordings[41,42] were applied in hippocampal granule cells to analyze the primary input of hippocampal circuit by stimulation of the lateral perforant pathway (LPP) (Fig. 5a). All cells showed a stable resting membrane

potential, with a mean value of −78 mV in both groups. First, we measured the rheobase, which reflects the minimum current required to generate an action potential (AP)[43]. ATR-FBΔ neurons had a decreased rheobase of about 40% compared to controls (control, 85 ± 9.8 pA vs. ATR-FBΔ, 53 ± 5.9 pA, $p = 0.007$) (Fig. 5b), indicating increased intrinsic excitability. To further test the excitability of ATR-FBΔ neurons and the ability to fire APs, we assessed and quantified the number of APs by applying 1000 ms current injections in graduated steps from 0 to 200pA. ATR-FBΔ neurons had a markedly higher number of APs ($p < 0.001$, Fig. 5c). In current-clamp train stimulation, ATR-FBΔ neurons displayed significantly higher AP amplitudes and longer AP half-width during 10 Hz (both $p < 0.001$, Supplementary Fig. 3d) and 20 Hz trains (both $p < 0.001$, Fig. 5d, e, f). Combined, these findings indicate that ATR-FBΔ neurons have altered intrinsic neuronal excitability, possibly underlying the lowered threshold for seizure initiation.

**ATR deletion enhances presynaptic activity and results in accumulation of presynaptic vesicles.** We proceeded to examine how ATR deletion affects the excitatory synaptic transmission between ATR-FBΔ neurons in the hippocampus—beginning by stimulating LPP afferents and measuring the amplitude and kinetics of AMPAR-mediated excitatory postsynaptic currents (EPSCs). ATR deletion effected no change on amplitude or

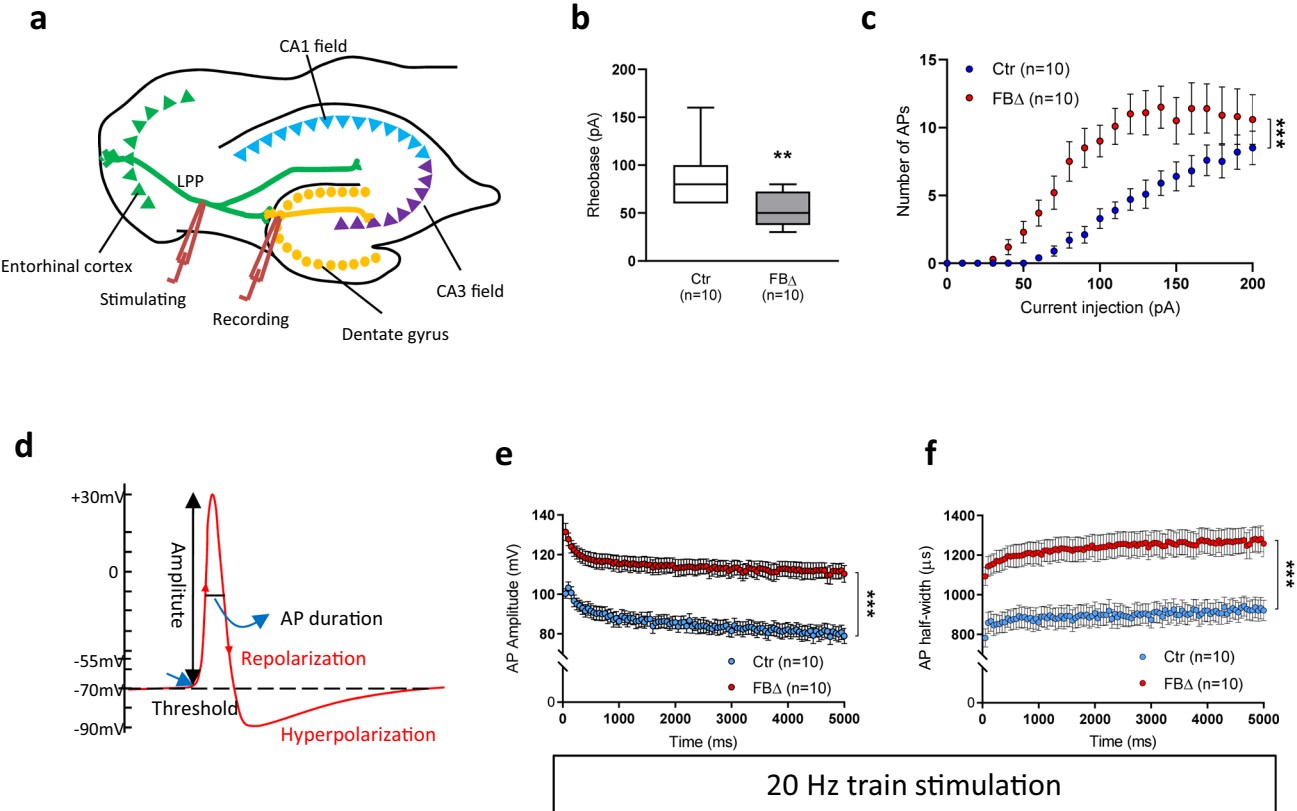

**Fig. 5 ATR-deleted neurons are more excitable. a** Schematic representation of hippocampal electrophysiology measurements on lateral perforant pathway (LPP)-dentate gyrus granule cell synapses. **b** Rheobase current of 1st provoked action potential by increasing stimulation in control (Ctr) and ATR-FBΔ (FBΔ) neurons. Box plot indicates the median (middle line), whiskers as max value (top) and min value (bottom) and hinges as 25 percentile (top) and 75 percentile (bottom). MWU test (unpaired, two-tailed), $p = 0.007$. **$p < 0.01$. **c** Number of action potentials (AP) in granule cells of the dentate gyrus in control ($n = 10$) and ATR-FBΔ neurons ($n = 10$) after 1000 ms current stimulation steps of 0 to 200 pA. Error bars indicate SEM. 2-way ANOVA with Holm-Sidak Post Hoc Analysis. ***$p < 0.001$. **d** Schematic representation of analysis on AP properties. **e** AP amplitudes within 20 Hz AP train. Error bars indicate SEM. 2-way ANOVA with Holm-Sidak Post Hoc Analysis. ***$p < 0.001$. **f** Plot of half-height width of AP in whole-cell recorded dentate gyrus granule cells within the 20 Hz train stimulation. Error bars indicate SEM. 2-way ANOVA with Holm-Sidak Post Hoc Analysis. ***$p < 0.001$. **b, c, e, f** The number of neurons measured is indicated within the graph. Source data are provided as a Source Data file.

kinetic parameters (Fig. 6a, b) of evoked EPSCs. To test the short-term synaptic plasticity, which depends on presynaptic neuro-transmitter release probability, we measured the paired-pulse facilitation (PPF) of LPP-granule cell (GC) synapses. The paired pulse ratio (PPR) was reduced in both 50 ms ($p = 0.009$) and 100 ms (albeit not statistically significant, $p = 0.077$) interstimulus intervals in ATR-FBΔ neurons (Fig. 6c, d), indicating an increased neurotransmitter release probability. These results indicate that ATR deletion particularly affects the presynaptic compartment and suggests enhanced vesicle release or vesicle content.

Transmission electron microscopy (TEM) analysis was per-formed on the dentate gyrus GC synapses of 3-month-old mice, so as to further differentiate the impact of ATR deletion on the

presynaptic changes. Remarkably, ATR-FBΔ neurons showed around 30% higher presynaptic vesicles (SVs) density than wildtype counterparts ($p < 0.001$) (Fig. 6e, f); although, the synaptic cleft width ($p = 0.94$) and length of the postsynaptic density ($p = 0.28$), as well as the post-synaptic density, were similar to controls (Fig. 6g, h). These data suggest that ATR deletion specifically disturbs homeostatic turnover of presynaptic SVs leading to accumulation of competent vesicles[44], without affecting postsynaptic sites or the active zone ultrastructure.

**Synaptic deficiencies in ATR-deleted neurons are independent from DDR.** ATR is a well-established DDR protein that safe-guards the genomic integrity from replication stress[4,5]. Western blotting was applied to examine whether the defects in neuronal/

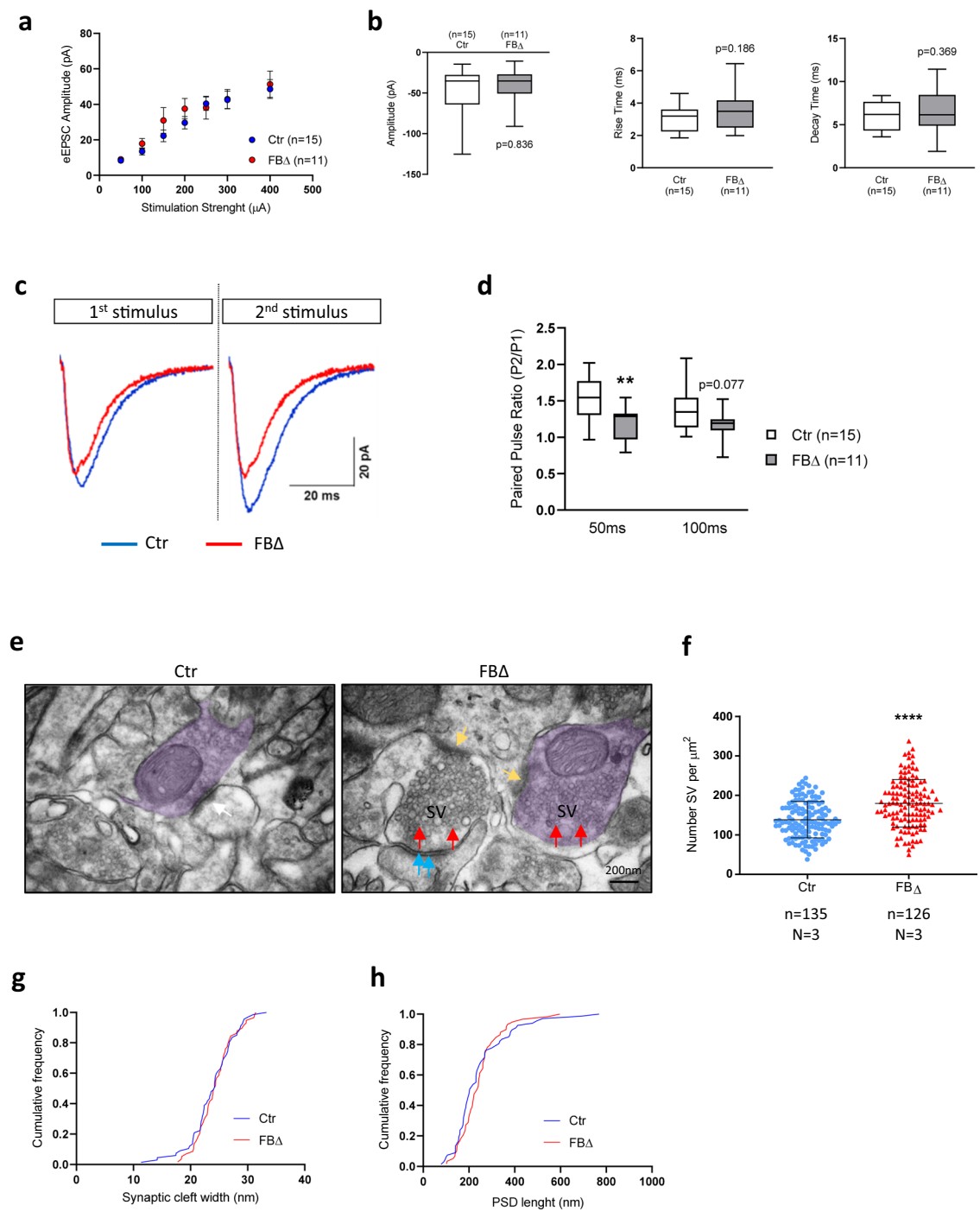

**Fig. 6 ATR regulates presynaptic activity and synapses homeostasis. a** Input–output (I–O) relationship of evoked excitatory post-synaptic currents (EPSCs) in granule cells of the dentate gyrus of control (Ctr) and ATR-FBΔ (FBΔ) genotypes at the indicated stimulus intensities. Error bars indicate SEM. 2-way ANOVA with Bonferroni $t$-test for individual stimulation steps ($p = 0.738$). The number of neurons measured is indicated within the graph. **b** Quantification of eEPSCs amplitude ($p = 0.836$), rise ($p = 0.186$) and decay time ($p = 0.369$) in granule cell-LPP of the dentate gyrus after supramaximal stimulation. The number of neurons measured is indicated under x axis. Error bars indicate SEM. Statistics by MWU test (unpaired, two-tailed) for amplitude, Student's $t$-test (unpaired, two-tailed) for rise and decay time analyses, respectively. Box plots indicate the median (middle line), whiskers as max value (top) and min value (bottom) and hinges as 25 percentile (top) and 75 percentile (bottom). The number of neurons measured is indicated within the graph. **c** Paired-pulse facilitation of eEPSCs in dentate gyrus granule cells. Sample traces of control ($n = 15$) and ATR-FBΔ neurons ($n = 11$) after supramaximal stimulation at 50 ms interpulse interval. **d** Quantification of Paired pulse ratio (PPR) after supramaximal stimulation at 50 ms ($p = 0.009$) and 100 ms interpulse intervals ($p = 0.077$). Box plot indicates the median (middle line), whiskers as max value (top) and min value (bottom) and hinges as 25 percentile (top) and 75 percentile (bottom). Student's $t$-test (unpaired, two-tailed). **$p < 0.01$. The number of neurons measured is indicated within the graph legend. **e** Representative electron microscopic pictures from the inner molecular layer of the dentate gyrus of 3-month-old control and ATR-FBΔ mice. The presynaptic bouton is colored in magenta and red arrows mark vesicles. Yellow arrows point to the active zone (synaptic cleft) and blue arrows to postsynaptic density. The images are representative from 3 mice per genotype analyzed. SV Synaptic Vesicles. **f** The number of SVs in control and ATR-FBΔ synapses. Data are presented as mean values ± SD. MWU test. ***$p < 0.001$. **g** Cumulative frequency diagram of synaptic cleft width (nm) in control and ATR-FBΔ synapses. **h** Cumulative frequency diagram of length of the postsynaptic density (nm) in control and ATR-FBΔ synapses. **f** The number of mice (N) and number of synapses (n) are shown. Source data are provided as a Source Data file.

presynaptic activity are secondary due to the ATR function in the DDR. First we analyzed a classical DNA damage marker poly (ADP-ribose) (PAR) in the ATR-FBΔ hippocampus, but found no detectable PAR formation in either ATR-FBΔ or control hippocampi (Fig. 7a). The DNA damage marker γH2AX signal was absent in both control and mutant hippocampi and ATR target phospho-Chk1 was low, but displayed no difference between mutant and control hippocampi (Fig. 7a). Of note, ATM expression was unchanged in mutants compared to controls (Fig. 7a). More importantly, the downstream substrates of ATM, such as Chk2 level or its active phosphorylated form (indicated by a supershift of Chk2), as well as phospho-p53, were very similar between mutants and controls (Fig. 7a). In addition, we detected no obvious increase of γH2AX signal in PCs of ATR-PCΔ brain sections in comparison with controls (Fig. 7b). Taken together, ATR deletion in postmitotic neurons neither induces DNA damages nor activates DDR signaling. Therefore, synaptic defects observed in ATR-FBΔ neurons are independent from DDR signaling.

**ATR targets SYT2 and PROT in post-mitotic neurons.** In searching for the molecular mechanism behind the ATR-mediated synaptic activities, we conducted quantitative proteomics analysis by mass spectrometry on 3-month-old hippocampi. This approach identified only 16 differentially abundant proteins (7 down-regulated, 9 upregulated, cutoff: $q$ value<0.25) in ATR-FBΔ hippocampi (Supplementary Fig. 4a). Of these, two presynaptic proteins—synaptotagmin 2 (SYT2) and sodium-dependent proline transporter (PROT)—attracted our attention, as both are shown to be involved in excitatory neurotransmission[45–47] and were the most affected in mutant hippocampi (Supplementary Fig. 4b). To validate the proteomic findings, synaptosome fractions were isolated from the hippocampal tissue and analyzed by Western blotting. We found that ATR was indeed localized in the synaptosome fraction and SYT2 and PROT were greatly elevated (around 5-fold and 2-fold, respectively) in synaptosomes of ATR-FBΔ hippocampi compared to controls (Fig. 8a). Of note, the SYT1 protein level, which is the homolog of SYT2 and the most studied member of the synaptotagmin family[46,48,49], remained unchanged (Fig. 8a). We find it interesting that no changes were observed in the expression of the postsynaptic proteins NMDA receptor subunit 2b (NR2B) and AMPA receptor subunits 1 (GluR1), as well as potassium channel 1.1 ($K_v$1.1) in synaptosomes of ATR-FBΔ hippocampi (Fig. 8a)—demonstrating a dispensable role for ATR in the postsynaptic compartment. As expected, the inhibitory marker GAD67 was unaffected since inhibitory neurons were not targeted by ATR deletion (Fig. 8a).

By means of further illustrating the bases on which ATR regulates SYT2 and PROT, we analyzed whether they interact directly. Neuroblastoma cells (N2a) were transfected with tomato-tagged SYT2 and PROT and immunoprecipitation (IP) was performed. We found that ectopically expressed SYT2 and PROT pulled down endogenous ATR (Fig. 8b, lanes 10, 12). The ATR inhibitor VE-821 did not influence their interactions (Fig. 8b, lanes 11, 13), indicating that ATR's kinase activity is not required for this interaction. Our interaction assay on endogenous proteins using hippocampal and cerebellar tissues confirmed the interaction of ATR with SYT2, as well as PROT (Fig. 8c). IP-Western blotting also detected an interaction of ATR with SYT1 in hippocampal extracts (Supplementary Fig. 4c). Taken together, these experiments indicate that ATR can interact with PROT and both SYTs and ATR depletion increased SYT2 and PROT levels in synaptosomes of the hippocampus.

**ATR deletion enriches association of SYT2 with excitatory neurons.** Despite their structural and functional homology, SYT1 and SYT2 have distinct expression patterns in the mouse brain[50] —the latter is preferentially expressed in inhibitory neurons, in contrast to SYT1 which expresses in excitatory neurons[45,51]. To visualize the location of SYT1 and SYT2, we co-stained the brain sections with SYT1 and SYT2 antibodies. Confirming previous reports, in control hippocampi SYT1 was localized in the hilar and inner molecular layer of the dentate gyrus and axonal regions, whereas SYT2 localized around the cell body of granule cells (Fig. 9a, b and Supplementary Fig. 5a) where the inhibitory contacts usually take place, but was absent in the hilus. Surprisingly, ATR deletion resulted in an aberrant enrichment of SYT2 in the hilus where the mossy fiber axons present, while its expression in the original location in granule cells was down-regulated, largely mirroring the pattern of SYT1 in wildtype controls (Fig. 9a, b and Supplementary Fig. 5a). Of note, SYT2 became colocalized with SYT1 in the mutant hilus (Fig. 9b). This change of expression and localization patterns by ATR deletion was also apparent in the CA3 region, where granule cell axons make their contact with excitatory pyramidal neurons (Supplementary Fig. 5b).

In order to further characterize those cell types expressing high SYT2 in the ATR-FBΔ hippocampus, we co-stained the brain sections with antibodies against SYT2 with either the inhibitory synapse marker VGAT, or the excitatory synapse marker VGLUT1. Normally, SYT2 predominantly colocalized with VGAT in granule cells and mossy fibers in the CA3 area in controls; however, in ATR-FBΔ hippocampi, SYT2 signals were less overlapping with VGAT in GCs and the mossy fibers in the

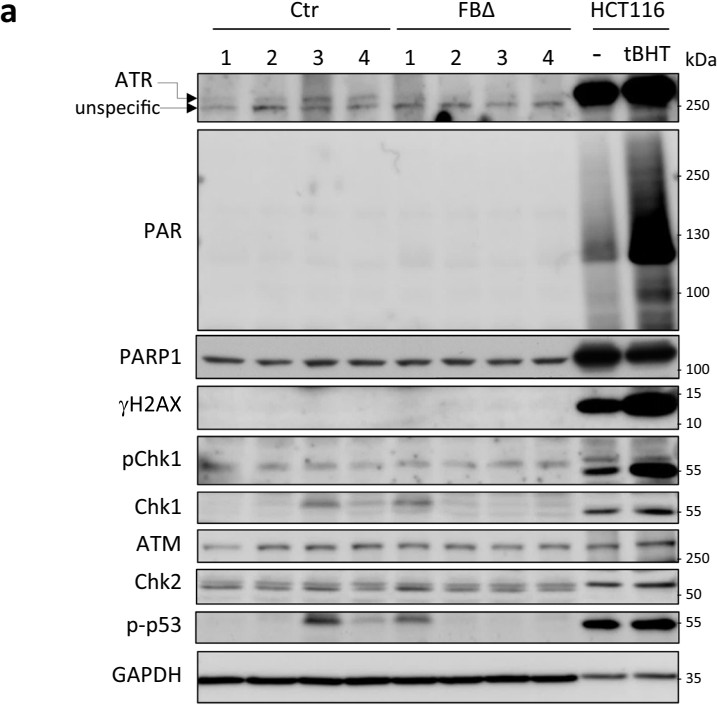

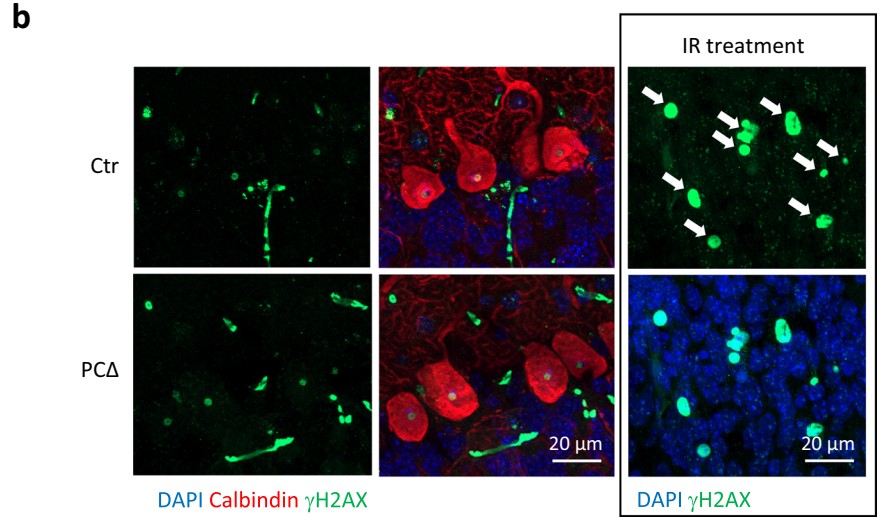

**Fig. 7 ATR deletion in neurons does not cause activation of the DDR. a** Western blot analysis of the hippocampus of 3-month-old control (Ctr) and ATR-FBΔ (FBΔ) mice for indicated proteins. The DDR activation is controlled by HCT116 cells without (−) or with (+) tBHP for 15 min. GAPDH serves as loading control. Four mice (numbers on the top of lanes) of the indicated genotypes were analyzed. **b** Immunostaining of ATR-PCΔ cerebellum with antibodies against Calbindin (for PCs) and γH2AX (DNA damage marker, white arrows). The γH2AX signal is controlled by the cortical sections of mice after 4 Gy ionizing irradiation (IR). The images are representative from 3 to 4 mice per genotype analyzed. Source data are provided as a Source Data file.

CA3 area (Fig. 9c, d and Supplementary 6a, b, c). In fact, SYT2 signals were particularly enriched in granule cell axons (mossy fibers) that are positive for VGLUT1 in both regions (Fig. 9e, f and Supplementary 6d, e). These observations strongly suggest that ATR deletion upregulates the availability of SYT2 in mossy fibers to facilitate vesicle fusion with the presynaptic membrane, rendering these neurons more excitatory.

## Discussion

Cells or tissues with a high replication capacity cannot tolerate ATR loss, since ATR is a key DDR molecule for handling replication stress[5,6]. Our study showed that when ATR is deleted from non-proliferating cells, for example in postmitotic neurons, it does not compromise the viability of these neurons and is dispensable for the formation and structure of the mouse brain. Both ATR-PCΔ and ATR-FBΔ mutant mouse strains have normal

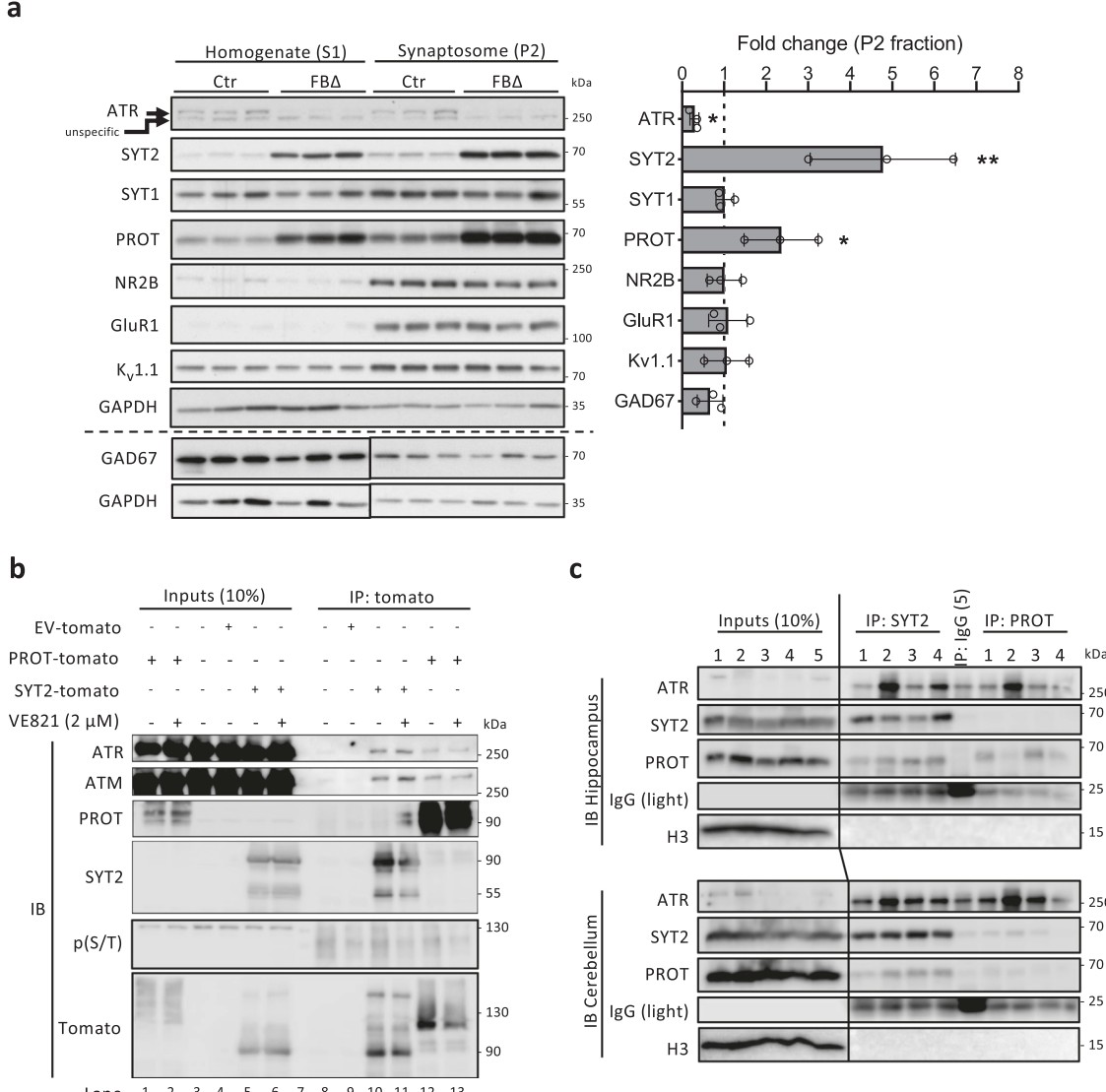

**Fig. 8 ATR deletion upregulates presynaptic proteins SYT2 and PROT. a** Western blot analysis of the indicated proteins in whole-cell extract (S1) and synaptosome (P2) fractions of 3-month-old hippocampus of control (Ctr) and ATR-FBΔ (FBΔ) mice. GAPDH controls loading. Note, GAD67 expression is controlled by a separate control GAPDH below. The right panel shows the quantification of the indicated proteins after normalization with GAPDH. The data is represented as relative fold change in protein expression in P2 fraction, compared to control mice. $N = 3$ mice for each genotype. The vertical dashed line in the graph indicates control mice. Data are mean values ± SD. Student's t-test (unpaired, one-tailed for ATR, SYT2 and PROT, two-tailed for others). P value: ATR, $p = 0.025$; SYT, $p = 0.001$; PROT, $p = 0.029$. *$p < 0.05$, **$p < 0.01$. **b** ATR interacts with SYT2 and PROT in vitro. Murine neuroblastoma cells (N2a) were transfected with Tomato-SYT2 or Tomato-PROT treated with or without with 2 µM ATR inhibitor VE-821. SYT2 and PROT were immunoprecipitated with Tomato antibody and blotted by indicated antibodies. The experiment was repeated four times. **c** Immunoprecipitation of protein extract from hippocampi and cerebella using antibodies as indicated. IP against IgG serves as control. Histone H3 serves as loading control for the inputs. IgG light chain was used to control the amount of antibodies used for IP. Note, an overloading of IgG in IP-IgG lanes correlates with unspecific binding signals in these samples. The number indicates individual mice. The experiment was repeated twice. Source data are provided as a Source Data file.

lifespan - in stark contrast to the severe neurodevelopmental defects and perinatal lethality of mice with ATR deletion in embryonic neuroprogenitors or other proliferating cells[11–13,52]. Also, other functions of ATR, such as protection against R-Loop-induced genomic stress or SSB repair[5,53,54] are likely dispensable for the life and the maintenance of these cells and neural tissues. Indeed, we detected neither obvious DDR activation nor apoptosis in these neurons. Once cells are spared replication stress, an abrogation of the ATR-DDR axis is neither toxic for the postmitotic neurons nor crucial for brain development.

Although ATR has been acclaimed to be an essential DDR regulator, our study demonstrates a previously unknown important and physiological function of ATR in neuronal activity.

Despite a normal morphology of the cerebellum and forebrain, ATR null neurons are defective in synaptic function. Interestingly, characterizing different neuronal phenotypes of both ATR-PCΔ and ATR-FBΔ mouse models allows us to test its function in inhibitory and excitatory neurons, respectively. Electrophysiological analyses of both ATR-PCΔ and ATR-FBΔ mouse models demonstrate an increased intrinsic neuronal activity of either inhibitory (PCs) or excitatory (hippocampal GCs) neurons. Thus, we conclude that ATR has a general physiological role in neuronal activity and synaptic function.

In PCs we found an increased firing rate without alterations of spiking regularity and spiking response after parallel fiber stimulation. These cellular abnormalities were associated with

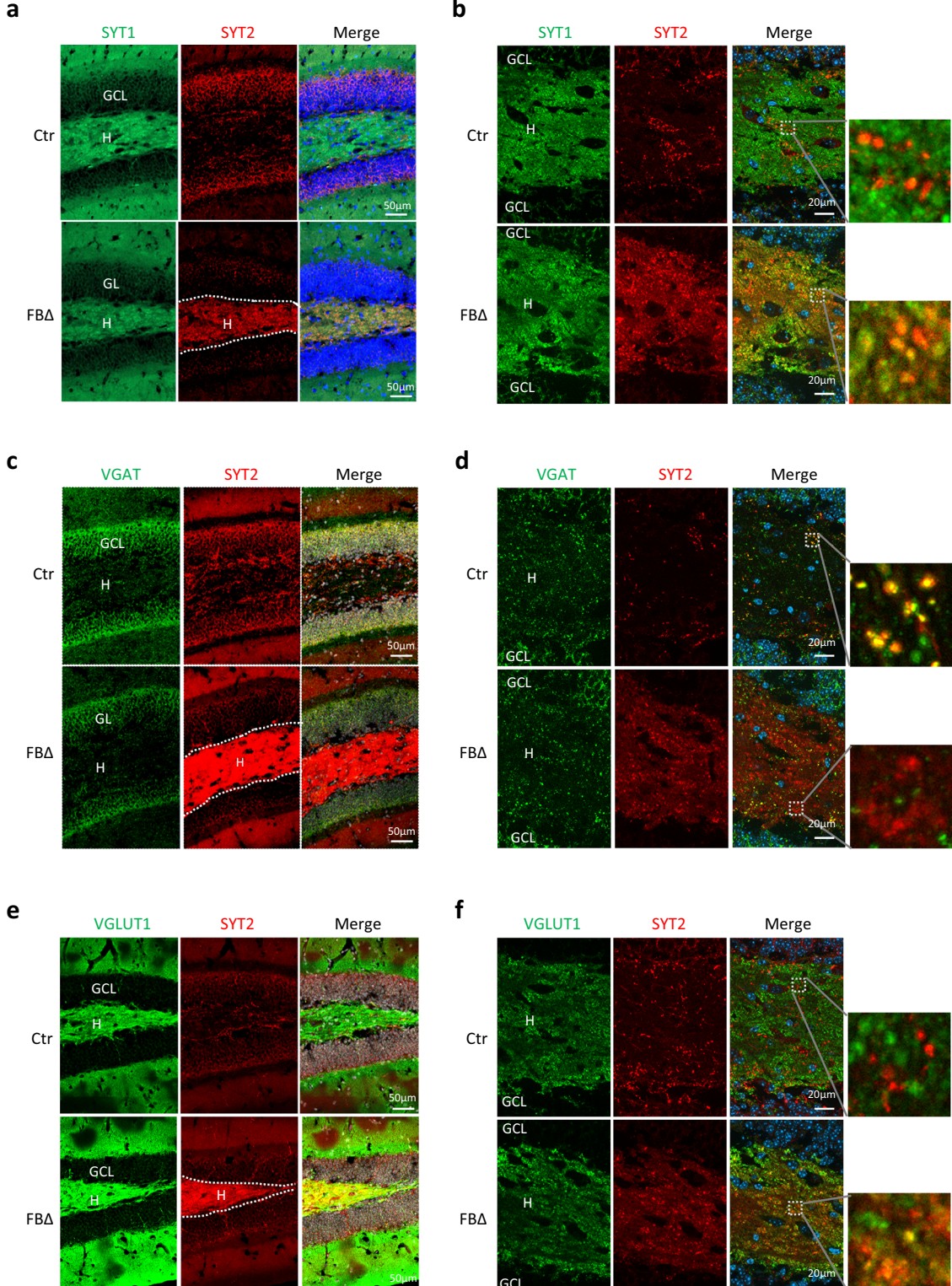

**Fig. 9 ATR deletion changes location and enrichment of SYT2. a** Sagittal sections from 3-month-old control (Ctr) and ATR-FBΔ (FBΔ) mice were stained with DAPI (blue), SYT1 (green) and SYT2 (red). The dentate gyrus area is shown. GL Granular Layer, H Hilus. **b** Immunofluorescence analysis of expression patterns of SYT1 and SYT2. Note a high SYT2 expression in granule cells in the GCL in controls, but more abundant and co-localization with SYT1 in mutant hilus (zoom-in pictures on right). **c** Sagittal sections from 3-month-old control and ATR-FBΔ mice were stained with DAPI (white), SYT2 (red) and VGAT (green). CA: Cornu Ammonis, DG: Dentate Gyrus). **d** Immunofluorescence analysis of expression pattern of VGAT and SYT2. Note a colocalization of VGAT and SYT2 in granule cells in the GCL in controls, but more abundant, yet not overlapping with VGAT, in mutant hilus (zoom-in pictures on right). **e** Sagittal sections from 3-month-old control and ATR-FBΔ mice were stained with DAPI (blue), SYT2 (red) and VGLUT1 (green). CA Cornu Ammonis, DG Dentate Gyrus. **f** Immunofluorescence analysis of expression pattern of VGLUT1 and SYT2. Note a high VGLUT1 in the hilus, where the SYT2 expression is low, in controls. Strong overexpression and co-localization (puncta) of VGLUT1 and SYT2 in the hilus (right zoom-in pictures). The images are representative from 4 to 5 mice per genotype analyzed.

age-dependent ataxic behavior of ATR-PCΔ mice. The firing rate depends on intrinsic excitability of PCs and the timing and interplay of parallel fiber and climbing fiber inputs onto PCs. The output from the PC to cerebellar nuclei is highly regulated to generate a system that ensures that the charge to the cerebellar nuclei neurons scales with the firing rate of the PCs. Of note, the impaired output of PCs could affect not only the cerebellar nuclei, but also directly neighboring PCs[55–57]. Alterations of PC firing rate have been found in multiple pathological conditions. In many genetic models of ataxia, e.g. of spinocerebellar ataxia, the spontaneous PC firing rate is reduced (see a recent review[58]) whereas in some other genetic, inflammatory, or stress models, PC firing rate is increased[59–62]. In general, the intrinsic excitability of PCs is influenced by voltage-gated sodium (VGNaC) and potassium channels (VGKC) and by $Ca^{2+}$-activated potassium channels (SK and BK channels). Specific alteration of ion channel function, e.g. blockade of D type $K_v1$channels[63,64] inhibits their control of PC hyperexcitability thus leading to increased PC firing. These notions may be compatible with the observed changes in AP anatomy of ATR-deleted forebrain neurons, such as larger AP amplitude (hyperpolarization) and an increased AP half-width (delayed repolarization), which may also indicate alterations of membrane bound cation channels[21]. AP shapes, amplitudes, hyperpolarization and inter-spike intervals, are all controlled by cation channels, such as VGNaC and VGKC. However, our proteomics did not detect any obvious changes of these channel proteins in ATR-deleted neurons. Although the interactome studies suggested an association of ATR with two sodium channel subunits, SCN2B and SCN3B in HEK293T cells[24,25], we found no change of these protein level (both proteomic and protein analyses) in ATR-deleted neurons.

While PCs' malfunction frequently affects ataxia, dystonia and tremor, interestingly, the cerebellum has also been identified as an initiator for epileptic seizures and targeted cerebellar modulation is currently evaluated for therapeutic seizure inhibition[65,66]. However, ATR-PCΔ mice do not show any seizure phenotype in the observation period of more than 20 months. Noticeably, while genetic disruption of many genes in PC mouse models display locomotor dysfunctions and ataxic deficits, some of which were associated with synaptic deficits and cerebellar atrophy, the seizure phenotypes have not been reported in those PC specific mutant mouse models (see review[67]). In addition, we recently reported that mice with specific deletion of HAT cofactor TRRAP (a life essential gene) specifically in PCs (TRRAP-PCΔ mice) showed an ataxia phenotype due to loss of PCs, but without seizures[68].

On the other hand, we observed spontaneously occurring generalized tonic-clonic seizures in the ATR-FBΔ mouse model. Consistently, we found increased population excitability in the CA3 region of the hippocampus as demonstrated by enhanced preictal activity. PEDs were increased in magnitude in ATR-FBΔ mice containing stronger slow (1–100 Hz) and nested fast ripple (200–500 Hz) oscillations, as well as increased spiking activity. These findings indicate an increased epileptogenicity in the hippocampal network[35,37,38] eventually underlying spontaneous seizures of ATR-FBΔ mice. Furthermore, we observed a prominent presynaptic defect and an increased neurotransmitter release probability without alterations of postsynaptic function possibly reflecting changes in neural mechanisms underpinning the population hyperexcitability. These defects are associated with a great accumulation of presynaptic vesicles, which likely account for altered neuronal network and AP properties[23]. Moreover, an aberrantly high spike frequency of ATR-deleted PCs and a decreased rheobase current of pyramidal neurons are indicative of increased excitability, which tallies with a higher AP number and synaptic vesicle accumulation in the presynaptic zone of ATR-

deleted pyramidal neurons. The maintenance of the membrane potential, rheobase and spiking behavior is crucial for neurotransmission, which otherwise leads to hyperexcitability and misbehavior, such as impaired cognitive functions, ataxia and epileptic encephalopathy[58,69]. Consistent with these notions, both ATR-PCΔ and ATR-FBΔ mice displayed behavioral deficits, i.e., locomotor coordination defects and epileptic seizures.

Strikingly, SYT2 and PROT1 are specifically upregulated and enriched at the ATR-FBΔ synapses. SYT2 is functionally and structurally related to SYT1[45]. Both proteins serve as an anchor to couple synaptic vesicles to the presynaptic membrane and, upon binding $Ca^{2+}$, trigger a rapid and synchronous release of neurotransmitters. Compared to SYT1, SYT2 is the faster $Ca^{2+}$ sensor and conducts a rapid release[46,70,71]. Although both are expressed in different areas of the hippocampus, ATR deletion enforced SYT2 co-expression together with SYT1 in the hilus. Intriguingly, we found a switch of the SYT2 expression profile from VGAT-positive to VGLUT1-positive synapses. It is thus plausible that ATR deletion added SYT2 to SYT1-expressing excitatory neurons, conferring upon them a rapid release kinetics.

PROT is a selective L-proline transporter that transfers proline from extracellular space into the presynapse and couples with synaptic populations in the basal state[72]. A high L-proline concentration can directly activate NMDA and AMPA receptors to enhance neuronal excitation and induce behavioral deficits like locomotor coordination defects and epilepsy[73,74]. It is worth noting that both PROT and SYT2 overexpression profiles are very similar and particularly upregulated in the synaptosome fractions of ATR-FBΔ hippocampi, suggesting that ATR deletion pushes SYT2 together with PROT in the same vesicle clusters of the presynaptic compartment[72,75]. Finally, ATR can interact with SYT2 and also PROT in vitro and in vivo, but specifically suppresses SYT2 and PROT, most likely depending on the cellular context and the type of synapses. For example, ATR sequester/ occupies SYT2 thereby rendering it less available to translocate normally to synaptic vesicles, whereas ATR depletion increases SYT2 and PROT levels in synaptosomes of the hippocampus. Alternatively, the interaction of ATR with SYT2 / PROT needs an adapter protein; the lack of ATR renders these proteins adopting a different conformation in the presynaptic compartment of excitatory neurons and execute specific downstream effects of these proteins in neuronal activities. It is also possible that ATR localizes with close vicinity to and interacts with SYT2 and PROT within synaptic vesicles, to prevent anchoring them onto presynaptic membranes. Its deletion relieves the repression of SYT2 and PROT in promoting membrane fusion to facilitate fast neurotransmitter release, therein rendering ATR-FBΔ mice more susceptible to epilepsy.

The change of the SYT2 expression pattern between VGAT+ and VGLUT1+ neurons is interesting. ATR has been reported to associate with inhibitory vesicle populations and interacts with VAMP2 and synapsin-1 in in vitro cultured neurons[26,76]. ATR deficiency has been shown to reduce inhibitory contacts, while increasing excitatory synapses in cultured neurons[76]. By way of contrast, our ATR-PCΔ and ATR-FBΔ mouse models rather indicate that ATR functions in both types of neurons. We currently do not have direct evidence on whether ATR deletion would accumulate a high density of presynaptic vesicles and confer a hyper-probability of presynaptic releases (or hyperexcitatory capacity) in PCs. Interestingly, we found also an interaction of ATR with SYT2 and PROT in cerebella and ATR deleted PCs show hyperactivation. However, whether ATR directly modulates presynaptic vesicle behaviors or function in PC neurons requires future investigation. Although the expression of ATR and ATM seemed to be reversely correlated in cultured neurons[76], we detected no changes of ATM expression

nor its DDR activity in ATR-deleted brains compared to controls, eliminating any compensatory role of ATM in specific type and functionality of neurons in vivo. Nevertheless, our study does not rule out a direct role of ATM in neuron activities and the synaptic compartment to tune neurotransmission as others reported (see a review[77]).

Our study shows that ATR functions to regulate the transmitter release probability/tendency, specifically in the presynaptic compartment, by modulating the distribution of fast $Ca^{2+}$ sensor SYT2 and the L-proline transporter PROT in presynaptic vesicles to couple with the presynaptic membrane (Supplementary Fig. 7). The involvement of ATR in the maintenance of intrinsic neuronal properties is not secondary to its classical DDR. The specific role of ATR in the presynaptic activity aids understanding of the neurological defects of ATR-SS and perhaps also other DDR disorders[78–80].

## Methods

**Mice**. $ATR^{flox}$, L7/pcp2-Cre, CamKII-Cre, ATR-PCΔ and ATR-FBΔ animals were generated, bred and housed in the mouse facility of Fritz Lipmann-Institute (FLI, Jena, Germany). Mice were fed *ad libitum* with standard laboratory chow and water in ventilated cages under a 12 h light/dark cycle. All animal experiments and breeding were conducted according to the German animal welfare legislation and approved by the Thüringer Landesverwaltungsamt (animal license nr.: 03-042/16).

*Rotarod performance test*. Before the testing period, mice were trained for one day on the rotarod apparatus (Ugo Basile, Italy) at a constant speed (5 rpm) for a maximum of 5 min, to allow them adapt to the testing environment. During the testing period, each mouse was placed on the rotarod at an accelerated speed (20 rpm/min), from 4 to 40 rpm for a maximum of 200 s. When the mice fell, they were removed from the apparatus and placed back into their cage for at least 30 min to recover before the next trial. All mice experienced three trials per day for five consecutive days. Latency to fall served as an indicator of motor coordination.

*Beam balance test*. For the beam walk balance test, 18–20-month-old female mice were initially trained for two days to run along firstly a 3 cm, then 2 cm width beam to their home cage. The test was performed on a 1 cm width beam for three trials per day for three consecutive days. The mice were videotaped and the time to cross the beam and the number of foot slips were measured.

### Cell culture

*Neuro2a (N2a) cell culture and transfection*. N2a cells (Neuroblastoma cells, ATCC® CCL-131) were cultured in a cell culture incubator at 37 °C in a 5% $CO_2$ and 20% $O_2$ atmosphere in culture medium containing DMEM (4.5 g/l glucose; Thermo Fisher), 10% fetal calf serum (FCS), 2 mM L-glutamine, 1 mM sodium pyruvate, 100 units/ml Penicillin, 100 μg/ml Streptomycin (all from Thermo Fisher). For transfection of N2a cells, $2 \times 10^5$ cells per well of 6-well plates were seeded and cultured overnight. Next day, Lipofectamine 2000 reagent (Invitrogen) was used to transfect the plasmids into the cells, according to the manufacturer's instructions. Transfected N2a cells were incubated for 24 h before harvesting.

*Primary neuron culture and transfection*. Primary hippocampal neuron isolations were performed from E18.5 mouse embryos. Briefly, hippocampi were collected into ice-cold 5 ml of 1x HBSS buffer containing 10% sucrose, then 1 ml of 0.25% trypsin was added for digestion at 37 °C for 15 min. The tissues were gently triturated with a glass Pasteur pipette to isolate the cells. $6 \times 10^4$ disassociated cells per 24-well plate were seeded in MEM medium (Thermo Fisher) supplemented with 1% glucose, 1x sodium pyruvate, 1% penicillin/streptomycin, 10 mM HEPES (Thermo Fisher), 2% B27 (Thermo Fisher), 1% FCS and 1 mM L-glutamine. The neurons were recovered for 1 to 3 h in a cell culture incubator at 37 °C in 5% $CO_2$ and 20% $O_2$. Then, the plating medium was replaced with Neurobasal Medium (Thermo Fisher) supplemented with 1% penicillin/streptomycin, 10 mM HEPES, 2% B27 and 0.5% GlutaMAX (Thermo Fisher). The neuron medium was refreshed every 3 days.

For transfection, neurons were cultured for 7 days and transfected with Lipofectamine 2000. Briefly, 15 ml of Neurobasal Medium (NBM) supplemented with 37.5 μl of 200 mM L-glutamine was used as incubation medium. Separately, 0.9 μg DNA and 1.65 μl Lipofectamine 2000 were diluted in 50 μl NBM per well of 24-well plate, mixed and incubated for 30 min at RT. The conditioned medium of the neurons was transferred to a new 24-well plate and kept in the 37 °C incubator. 500 μl of incubation medium and 100 μl of DNA + Lipofectamine mixture were added to the neurons and incubated at 37 °C for 45 min. The transfection mixture was then replaced with the conditioned medium. Transfected neurons were fixed with 4% PFA/sucrose and subjected to immunofluorescence staining 7 to 10 days after transfection.

**Total protein extraction from mouse brain**. The isolated hippocampal tissues were homogenized with Precellys 24 Tissue Homogenizer (Program settings, 4000 rpm 3 × 10–5 s) in 500 μl of RIPA buffer (50 mM Tris/HCl (pH 8.0), 150 mM NaCl, 1 mM EDTA, 1% Triton X-100, 1% sodium deoxycholate and 0.1% SDS) supplemented with PhosphoStop® (Roche) and protease inhibitor cocktail (Roche). The homogenates were incubated on ice for 45 min, followed by brief sonication in a Bioruptor (Diagenode) at 4 °C (5 cycles of 30 s on and 30 s off). The sonicated homogenate was centrifuged at full speed for 20 min to clear up the protein fraction. Protein quantification was performed with Pierce™ BCA Protein Assay Kit (Thermo Fisher Scientific), following the manufacturer's instruction.

**Synaptosome extraction from mouse brain**. To isolate the synaptic protein fraction from the hippocampus, Syn-PER™ Synaptic Protein Extraction Reagent (Thermo Fisher Scientific) was used according to the manufacturer's instructions. Briefly, the freshly isolated hippocampal tissue was gently homogenized with a Dounce tissue grinder on ice in 500 μl of Syn-PER™ reagent, supplemented with PhosphoStop® (Roche) and protease inhibitor cocktails (Roche). The homogenate was centrifuged at 1200 g for 10 min at 4 °C. The supernatant was centrifuged again at 15,000 g for 20 min at 4 °C. The resultant synaptosomal pellet (P2) was washed once and resuspended in Syn-PER™ reagent.

**Immunoprecipitation (IP)**. After harvesting by scraping, N2a cells were lysed in NET-N buffer (20 mM Tris HCl pH 8.0, 137 mM NaCl, 10% glycerol, 1% NP-40, 2 mM EDTA), followed by brief sonication in a Bioruptor (Diagenode) at 4 °C (5 cycles of 30 s on and 30 s off). Meanwhile, the Dynabeads (Invitrogen) were washed 3 times and resuspended in NET-N buffer. 10% of the inputs from cell lysates were retained for immunoblotting. The beads were mixed with cell lysates and 2 μg of tdTomato polyclonal antibody (SICGEN) per IP. IP was performed at 4 °C for 2 h with gentle rotation. Following incubation, the beads were washed 3 times with NET-N buffer and boiled at 95 °C for 10 min in 2×SDS loading buffer. The IPs from the mouse brains lysates were performed as previously described[81].

**SDS-PAGE and western blot analysis**. The protein samples were diluted in SDS-sample buffer and boiled at 95 °C for 5 min. 10–20 mg of protein per lane were run on 7.5–12.5% SDS-PAGE polyacrylamide gels and transferred onto nitrocellulose membranes (Bio-Rad). After blocking in 5% milk-TBST, the membranes were incubated overnight at 4 °C with primary antibodies and visualized with a chemiluminescent system (Thermo Fisher Scientific.) Protein expression levels were quantified using ImageJ.

The following antibodies and dilutions were used in the study: ATR 1:250 (SantaCruz, #sc-515173), GAPDH 1:5000 (Sigma-Aldrich, #G8795), ATM 1:1000 (Novus Biologicals, # NB100-309), PAR 1:1000 (Trevigen, # 4336-BPC-100), PARP1 1:1000 (SantaCruz, #sc-1561), Chk1 1:1000 (Bethyl Laboratories, #A300-162A), phospho-Chk1 1:1000 (Bethyl Laboratories, #A300-163A), γH2AX S139 1:1000 (Upstate, #05-636), Syt1 1:1000 (Synaptic Systems, #105011), Syt2 1:1000 (Synaptic Systems, #105123), Slc6a7 1:400 (GeneTex, #GTX65943), NR2B 1:1000 (Millipore, #06-600), GluR1 1:1000 (SantaCruz, #sc-13152), $K_v1.1$ 1:200 (Alomone, #APC-009), GAD67 1:1000 (Abcam, #ab26116), phospho-(S/T) ATM/ATR substrate 1:1000 (Cell Signaling, #2851 S), tdTomato 1:1000 (SICGEN, #AB8181-200), H3 1:1000 (Abcam #ab1791). Detection of IgG light chain by secondary antibodies (1:5000 in 2.5% milk-TBS-T). Detection of specific signals for Syt2 and Prot1 with Veriblot secondary antibodies (1:2500 Abcam #ab131366 in 2.5% milk-TBS-T). Full scanned images of Western blots are provided in "Source Data File".

**Histology**. Transcardial perfusion of the whole mouse and dissection of the fixed brains was performed and isolated brains were post-fixed in 4% PFA fixative solution at 4 °C for overnight and cryoprotected in 30% sucrose solution. 20 μm sections were prepared and used for immunofluorescence or Nissl staining.

**Immunofluorescence staining of brain sections and TUNEL reaction**. For TUNEL reaction and immunofluorescence staining, the sections, after antigen retrieval, were blocked in a blocking solution (5% donkey serum, 1% BSA, 0.4% Triton X-100 in PBS) and incubated overnight at 4 °C with primary antibodies prepared in the blocking solution. Next day, the sections were incubated with the secondary antibodies prepared in blocking solution for 1–2 h. After several washes in PBS, DNA was counterstained with DAPI. Slides were mounted with Pro-Long™ Gold Antifade Mountant (Thermo Fisher Scientific). Images were acquired with Zeiss Axio Imager-ApoTome Axiovert 200 ApoTome.

The following antibodies and dilutions were used for immunofluorescence stainings: calbindin D28K 1:1000 (SantaCruz, #7691), GFAP 1:4000 (Dako, #Z0334), myelin binding protein 1:200 (Millipore, #MAB384), GM130 1:400 (BD Biosciences, #610822), GFP 1:200 (SantaCruz, #sc-9996), tdTomato 1:200 (SICGEN, #AB8181-200), NeuN 1:4000 (Millipore, #MAB377), SMI312 1:4000 (Covance, #SMI312-R), Znt3 1:200 (Synaptic Systems, #197002), Syt1 1:200 (Synaptic Systems, #105011), Syt2 1:200 (Synaptic Systems, #105123), VGAT 1:200 (Synaptic Systems, #131003), VGLUT1 1:200 (Synaptic Systems, #135303), Streptavidin-Cy3 1:800 (Sigma, #S6402-1ml), anti-mouse Cy3 1:200 (Sigma, #C2181-1Ml), anti-rabbit Cy3 1:200 (Sigma, #C2306-1Ml), anti-rabbit Cy2 1:200 (Jackson Immuno Research, #711-225-152), anti-rabbit Cy5 1:200 (Jackson

Immuno Research, #711-175-152), Alexa Fluor 488 donkey anti-Mouse igg (H + L) 1:200 (Invitrogen, #A21202).

**Transmission electron microscopy (TEM)**. For TEM studies 3-month-old animals were perfused intracardially with cold EM fixative (3% glutaraldehyde, 1% paraformaldehyde, 0,5% acrolein, 4% sucrose, 0,05 M CaCl₂ in 0,1 M cacodylate buffer, pH 7.3). The hippocampus was isolated and postfixed for at least one day. For secondary fixation, the samples were incubated in 2% OsO₄/1% Potassium ferrocyanide in 0,1 M cacodylate buffer for 3 h at 4 °C in the dark, followed by dehydration in an ascending water/acetone series and embedded in epoxy resin 'Epon' (glycid ether 100, SERVA). The resin was allowed to polymerize for 2 days at 60 °C in flat embedding molds. After curing, the samples were trimmed with a Reichert UltraTrim (Leica). Semithin sections of 0,5 µm with an Azure staining[82] were executed to allow an orientated survey of the hippocampus formation in LM. Ultrathin sections of 55 nm without post staining were placed onto copper slot grids coated with a Formvar/Carbon layer, for TEM analysis. All sections were made with an ultramicrotome (Reichert Ultracut S; Leica, Wetzlar, Germany) and electron micrographs were taken on a JEM 1400 electron microscope (JEOL, Japan), using an accelerating voltage of 80 kV and coupled with Orius SC 1000 CCD-camera (GATAN, USA).

## Electrophysiology

*Cerebellar recordings.* The brains of 18–20-month-old mice were removed in ice cold protective cutting artificial cerebrospinal fluid (aCSF1) containing (in mM): 95 N-Methyl-D-glucamine, 30 NaHCO₃, 20 HEPES, 25 glucose, 2.5 KCl, 1.25 NaH₂PO₄, 2 thiourea, 5 sodium ascorbate, 3.0 sodium pyruvate, 10 MgSO₄, 0.5 CaCl₂, 12 N-acetylcysteine, adjusted to pH 7.3 and an osmolarity of 300–310 mOsmol, saturated with 95% O₂/5% CO₂[83] and cut into halves. 300 µm thick coronal slices were made from the cerebellum with a vibratome (VT1200S; Leica, Wezlar, Germany). Slices were placed in an incubation beaker with aCSF1 at 34 °C for 10–15 min, then transferred into another incubation beaker with aCSF2 (in mM): 125 NaCl, 25 NaHCO₃, 25 glucose, 2.5 KCl, 1.25 NaH₂PO₄, 1 MgCl₂, 2 CaCl₂, 2 thiourea, 5 sodium ascorbate, 3 sodium pyruvate, 12 N-acetylcysteine, adjusted to pH 7.3 and an osmolarity of 300–310 mOsmol, saturated with 95% O₂/5% CO₂) until use.

Loose-Patch recordings from cerebellar Purkinje cells (PCs) were performed in aCSF3 containing (in mM): 125 NaCl, 25 NaHCO₃, 25 glucose, 2.5 KCl, 1.25 NaH₂PO₄, 1 MgCl₂, 2 CaCl₂, saturated with 95% O₂/5% CO₂) as described previously[84], using thick-walled borosilicate glass recording electrodes (2.0 mm o.d., Science Products, Germany) filled with aCSF3 and a final resistance of 3–5 MOhm. PC spikes were recorded for 100 s. Evoked responses were elicited by stimulating parallel fibers about 200 µm away from the PC bodies using a monopolar stimulation with an additional pipette containing aCSF3 (DS3, Digitimer). Frequency, interspike interval and CV of ISI were analyzed with Neuromatic Plugin of IgorPro software (WaveMetrics, Portland, OR, USA).

*Hippocampal patch-clamp recordings.* After decapitation of the 3-month-old mice, the brain was isolated in ice-cold protective aCSF1. 300 µm coronal hippocampal slices were prepared. The slices were recovered in aCSF1 at 34 °C for 10–15 min, then incubated with aCSF2 at RT for at least 1 h until recording. Electrophysiological measurements were conducted with a HEKA EPC-10 patch-clamp amplifier with a sampling rate of 20 kHz. All recordings were filtered at 2.9 kHz using the amplifier's Bessel filters. For whole-cell patch-clamp recordings, single slices were transferred into a recording chamber and continuously submerged with aCSF3. Dentate gyrus granule cells were held at −70 mV during recording. Patch pipettes (2.5–5 MΩ) were pulled from borosilicate glass and pipette intracellular solution contained (in mM): 120 KGluc, 20 KCl, 10 HEPES, 0.1 EGTA, 4 Mg-ATP, 0.2 Na₂-GTP, 2 MgCl₂, 10 Na-Phosphocreatine, pH 7.3, 280 mOsmol. Series resistance was compensated (70–80 %) and cells with series resistance >30 MΩ or series resistance changes of >20% during measurement were discarded. For evaluation of evoked excitatory postsynaptic currents (eEPSCs), the LPP was stimulated by a bipolar theta glass electrode filled with aCSF3 and connected to a stimulus isolation unit (Isoflex, A.M.P.I, Jerusalem, Israel). Supramaximal stimulation was determined when increasing stimulation did not result in an increase of eEPSC and ranged from 200 to 400 µA. AMPAR-mediated glutamatergic events were isolated by applying 50 µM 2-amino-5-phosphonovalerate (AP-5, Tocris) and 20 µM bicuculline (Sigma-Aldrich) in aCSF3 bath solution. Paired-pulse facilitation of eEPSC was measured with interstimulus intervals of 50 ms and 100 ms after supramaximal stimulation. Whole-cell recordings were analyzed by NeuroMatic plugin of Igor Pro software 6 or 7 (Wavemetrics).

*Induction of epileptiform activity and recording of local field potentials (LFP) in hippocampal CA3.* Transversal hippocampal slices (400 µm thick) from 3-month-old mice (4 control and 4 ATR-FBΔ) were prepared in ice-cold aCSF1, recovered for 10–15 min in aCSF1 at 34 °C and were incubated for one hour in aCSF2 at room temperature. Afterwards the slices were stored in a beaker with aCSF4 containing (in mM): 125 NaCl, 25 NaHCO₃, 10 glucose, 2.5 KCl, 1.25 NaH₂PO₄, 1,5 MgSO₄, 2 CaCl₂, adjusted to pH 7.3 and an osmolarity of 290–300 mOsmol, saturated with 95% O₂/5% CO₂) at room temperature until recording. Slices were

transferred in a custom-made interface recording chamber placed on an inverted microscope and were perfused (3–4 ml/min) with warm (31–34 °C) Mg-free aCSF5 with a slightly elevated K + concentration to induce epileptiform activity (in mM: 125 NaCl, 25 NaHCO₃, 10 glucose, 4 KCl, 1.25 NaH₂PO₄, 2 CaCl₂, adjusted to pH 7.3 and an osmolarity of 290–300 mOsmol, saturated with 95% O₂/5% CO₂.) Recording electrodes consisting of borosilicate micropipettes filled with Mg-free aCSF5 (2–4 mOhm) were placed into the pyramidal layer of CA3c-b. Recordings of LFP were started 10 min after transfer of the slice to the epileptogenic Mg-free aCSF5 to standardize the onset of ictal activity. 40-minute recordings were acquired at 5 kHz sampling rate and were low-pass filtered at 2 kHz with a Butterworth filter using a HEKA EPC-10 patch-clamp amplifier (HEKA Elektronik GmbH).

*LFP data analysis.* Extracellular field potential signals were corrected for the baseline using a moving-median of 2 s time window. We identified possible artifacts by visual inspection and excluded from the signals by considering them as missing values. The 50 Hz power line interference artifact and its harmonics were removed using notch filter (*iirnotch* and *filitfilt* functions of MATLAB). Each signal was then digitally filtered to obtain (1) population field activity (PFA; 1–100 Hz band-pass FIR filter), (2) fast ripple activity (FRA; 200–500 Hz band-pass FIR filter), and (3) multiple unit activity (MUA; 500 Hz high-pass FIR filter) signals. The filtering was performed using *bandpass* and *highpass* functions of MATLAB.

Preictal epileptiform discharges (PEDs) were detected based on the PFA signals (1–100 Hz). To this end, we computed the local variability level in each signal using a moving-standard-deviation of a 75 ms time window (*movstd* function of MATLAB). This was then subjected to a general-purpose event detection routine in order to detect the PEDs[85], followed by a visual verification of the detected events. For this analysis, the leading threshold parameter of the detection algorithm was set to 3 times the SD of the noise level. To compare the activity characteristics of PEDs to that of the baseline we computed a non-PED time-series for each signal. To this end, we excluded PEDs by considering 150 ms before and 300 ms after each event (i.e. a time window of [−150 300] ms). These excluded periods were considered as missing values. Using this relatively extended time window yielded a robust exclusion of PEDs in the non-PED signal.

Power spectral density (PSD) analysis of PEDs followed in time windows set at [−150 150] ms centered on the PED peak time. We computed the Fourier transform (FFT; *periodogram* function of MATLAB) of each event in PFA signal (1–100 Hz), and averaged across the events. As the baseline PSD, we computed the FFT of 300 ms epochs of the corresponding artifact-free non-PED signal, and averaged across the epochs. For testing the difference in PSD across groups (Control vs. FBΔ), we accounted for the 1/f power scaling using the decibel baseline normalization[86]. To this end, we divided the PSD of PEDs at each frequency by that of the non-PED spectrum, and computed the 10 times of its logarithm with base 10. We then averaged the normalized PSD over 1–100 Hz to obtain the average normalized power in this band. For gamma band, we averaged over 30–90 Hz. By applying the same approach to FRA signals, we computed the average normalized power in 200–500 Hz.

To investigate the evolution of strength of population field activity over time we computed the total instantaneous power of the signal in 1–100 Hz band. To this end, we first computed the instantaneous power of the PFA signal at each time point by using its analytic signal resulted from the Hilbert transform (*hilbert* function of MATLAB). This enabled computing the maximum instantaneous power of each PED. We then partitioned the recording time (~40 min) to the non-overlapping bins of 5 min, and computed the average of these maximum powers of the PEDs within each bin. The resulted time-series represents the total instantaneous power of the PEDs within each 5 min. For testing the difference in the total instantaneous power between the two groups (Control vs. FBΔ) at each bin, we used the one-tailed permutation test with 1,000,000 times shuffling at a significant level of 5% while accounting for the multiple-comparison problem by the method of Cohen[86].

To detect the spikes within PEDs we considered the MUA signal in a window of [−50 150] ms centered on the peak time of each PED. In our preliminary analysis, we found that this window is at the same time sufficiently wide to cover the main activity phase of each PED, and also useful for increasing the robustness of the spike detection against the potential false-positive errors. For this analysis, we only accepted the spikes with amplitude greater than three times the root mean square of the baseline (i.e. non-PED) signal[35]. The detected spikes were verified visually.

The abovementioned analyses of the filed potential data were performed in MATLAB 2020 (MathWorks, Natick, MA, USA) using custom written code, the publicly available toolbox UFARSA[85] and MATLAB's built-in functions.

**Protein mass spectrometry (proteomics)**. The hippocampi of 3-month-old animals were isolated (4 mice per genotype), weighed and snap frozen in liquid nitrogen. Sample preparation, TMT labeling, data acquisition and data processing for Mass Spectrometry (MS) was performed as previously described[87,88].

**Statistics**. Statistical analyses were performed using GraphPad Prism 8, Sigmaplot 13 and MATLAB 2020. First, the data set were subjected to Shapiro-Wilk test to check for the normal distribution. If the data set passed the Shapiro–Wilk test (*p* value>0.05), Student's *t* test was used to determine the statistical significance. If the data set did not pass the Shapiro–Wilk test (*p* value<0.05), the statistical

significance was determined by Mann–Whitney *U* test. One-way-ANOVA was used for the comparison of more than one group. The exact *p* values are provided, unless it is <0.001, and *p* < 0.05 was accepted as the level of significance for all of the tests.

**Reporting summary**. Further information on research design is available in the Nature Research Reporting Summary linked to this article.

## Data availability

The mass spectrometry proteomics data have been deposited to the ProteomeXchange Consortium (http://proteomecentral.proteomexchange.org)[89–91] via the PRIDE partner repository[92] with the ProteomeXchange accession identifier PXD013414 (ATR-FBΔ hippocampus tissue). Source data are provided with this paper.

## Code availability

The custom code for the LFP data analysis is available from the corresponding author on request. This code was generated using MATLAB 2020 built-in functions (as mentioned in Methods) and an event detection routine (UFARSA) which is publicly available at (https://github.com/VahidRahmati/UFARSA).

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

## Acknowledgements

We thank P. Elsner for his excellent assistance in the maintenance of the animal colonies. We also thank the FLI core facilities of imaging, histology and proteomics, in particular Erika K. Sacramento and Norman Rahnis, for their excellent services. We are grateful to E. Brown (University of Pennsylvania, USA) for sharing ATR floxed mice. We are also grateful to M. Halilovic, who assisted in sample preparation. We thank members of Wang laboratory for their critical and helpful discussions. Figure 2a was created with BioRender.com. We thank L. Witter for his thoughtful discussion. This project is supported by DFG grants (to Z.-Q.W., WA2627/1-1, WA2627/5-1, to C. G., GE2519/8-1, GE2519/9-1) Germany, grants from the Leibniz Association (to Z.-Q.W., SAW2014, SAW2015), a grant from the German-Israel Foundation (GIF) (to Z.-Q.W. I-1307-418.13/2015) and the German Ministry of Education and Research (BMBF) (to C.G. 01EW1901, 01GM1908B), and by the Schilling Foundation (to C.G.).

## Author contributions

M.K. conceived the project, performed the majority of the experiments, interpreted the data and wrote the manuscript. C.M. performed protein assays. J.S., H.H., M.C., and V.R. performed electrophysiological analyses. Z.-W.Z. generated mouse models, designed and performed behavioral experiments, characterized mouse phenotype and interpreted the data. A.O. performed proteomic analysis. K.B. performed TEM. P.G. contributed to the imaging data acquisition. C.G. supervised electrophysiological analyses, contributed to the project development and wrote the manuscript. Z.-Q.W. designed the experiments, supervised the project and wrote the manuscript.

## Funding

## Competing interests

The authors declare no competing interests.
