## [Peer Review File · Nature Communications]

Reviewers' Comments:

Reviewer #1:

Remarks to the Author:

In this paper Wang and colleagues model ATR deficiency in mice by generating cell and tissue specific deletions. This is important because hypomorphic mutations in ATR cause growth retardation, microcephaly and intellectual disability which suggests a role for ATR in non-dividing neurons, yet this role remains unknown. The authors describe a series of elegantly designed experiments to address this question, which led to a new concept - ATR deletion in post-mitotic neurons does not impact neuronal cell survival but rather causes aberrant firing leading to ataxia and epilepsy. The authors then employ proteomic and co-immunoprecipitation assays to show an interaction between ATR and synaptic proteins SYT2 and PROT. They further show that ATR deletions result in upregulation of SYT2 which is then aberrantly translocated to excitatory neurons, providing a mechanistic explanation of the in vivo observations. Overall the manuscript is well-written, the conclusions are novel and appropriately supported by data, which provides a new concept in the field that will be of broad interest to the readers of Nature communications. However, there are a few points that need to be addressed before the manuscript is accepted.

1- The interaction of ATR with SYT2 and PROT is kinase independent but appears to elicit a preferential impact on excitatory neurons. Does ATR adopt a different conformation in excitatory neurons compared to others or is there an adaptor protein that mediates downstream effects of the interaction, which is only present in excitatory neurons? What underlies the cell-specific consequences of the interaction? This might be difficult to address experimentally but the reader will benefit from further elaboration on this point as it is critical to the overall model in the manuscript.

2- How does absence of ATR upregulate SYT2 and cause aberrant translocation, which is again kinase independent? Is it likely that ATR sequester SYT2 thereby rendering it less available to translocate? In follow-up studies, it would be interesting to map the domain in ATR that interacts with SYT2 and use deletion mutants to confirm the model.

3- Once replication stress is spared, ATR deficiency is not toxic to neurons nor required for brain development but is required for physiological synaptic function. The authors rule out a compensatory role for ATM – Does this imply that ATM is not required for synaptic function?

4- Are there known ATR variants (or SNPs) that disrupt the interaction with SYT2 and is there evidence for association with neurological disease?

Reviewer #2:

Remarks to the Author:

In this manuscript, the authors use a combination of mouse genetics, slice electrophysiology, and behavioral analyses to test the role of ATR in the brain. Previous studies fail to address the adult neural role of ATR because straight knock out mice are not viable. Therefore, using a conditional approach, they were able to look at GABAergic neuron function in the cerebellum and excitatory neuron function in the hippocampus. The authors show that loss of ATR in neurons causes changes in neuronal excitability, and the mice have an increased susceptibility for ataxia and epilepsy. The authors show that the deletion of ATR affects the presynaptic compartment, where it interacts with synaptotagmin 2 and PROT. The paper is generally well written and easy to follow. The images are very nice. The use of conditional genetics is a major plus. Despite the potentially broad interest of this paper, there are a number of issues that lower enthusiasm. In my comments, I outline the major problems that I hope will help the authors improve the paper.

1) The authors indicate in the title that ATR regulates neuronal activity in vivo. Although I somewhat understand what the authors generally mean here, I think it would be fairer to eliminate the in vivo part. No in vivo recordings were actually conducted and this is exactly what someone reading this paper would assume from the title.

- 2) The authors have indicated in several instances "data not shown". This is unnecessary as these data actually seem very important and should be included.
- 3) Related to the above, the whole story and depth of analysis is rather superficial. For example, the cerebellar electrophysiology essentially teaches us that the overall activity of Purkinje cells is affected. Given the use of slice electrophysiology, I would have expected a deeper analysis of Purkinje cell function. In this regard, it's unclear what the authors' findings mean in the larger context. There have now been many slice and in vivo cerebellar recording studies showing how changes in Purkinje cell firing rate and pattern underlie different models of ataxia, dystonia and tremor. My question is, how do the current results fit into those models?
- 4) Perhaps the most interesting part of the paper is that the mice develop seizures. However, unless I have missed it, I do not see any EEG recording or even videos showing the seizures. There is much work needed in order to appreciate what this phenotype looks like and what it actually means. For instance, what neurons/brain regions are involved in the seizures and what specific neuronal defects are driving the seizures? More generally, although the idea of these mice having seizures is exciting, the rationale behind this part of the paper and the actual goal of looking at seizures was underdeveloped. There is an exciting drive in the field showing that the cerebellum might be a key region initiating seizure activity. It would have been nice to see such papers cited and discussed.
- 5) I am not sure what the authors are referring to by "motor strength".
- 6) The authors state "...to avoid the secondary effects of epileptic seizures...". This is confusing since seizures were a part of the phenotype. But here the authors wish to avoid them. There is no rationale provided for this.
- 7) Neuroblastoma cells were used for the protein-protein interaction studies. What happens in vivo? And specifically, what do these interactions look like in the cerebellum versus the hippocampus?
- 8) Overall, the reason for a split of the electrophysiology into cerebellar and hippocampal sections is unclear. Each system is given a somewhat superficial treatment and as a result I am really unsure what we have learned. Why not dig into one of these systems a little deeper? Again, please see my above comment about other models of cerebellar function in ataxia. We know so much about this topic and little of the prior knowledge is exploited here to advance the understanding of ataxia or cerebellar function.
- 9) The Discussion section is lacking many essential arguments. For instance, there is no discussion about the seizure phenotype, no discussion about what the cerebellar dysfunction means or any potential mechanisms therein, and there is no discussion about how the synaptic and molecular defects potentially drive cerebellar-dependent motor behaviors.
- 10) In Figure 1B, it's not so convincing to me that there are no cerebellar defects. Subtle as it might be, I see defective Bergmann glia processes. Also, for both the control and the mutant, the calbindin staining could be better quality.
- 11) In Figure 1, it's peculiar to me that the authors chose to examine lobule X. This lobule is well known to have a neuroprotective nature in many different diseases (see the older work from Richard Hawkes, but you can also see the newer work from Zoghbi, Shakkottai, and other labs). Thus, perhaps the authors need to look at other lobules as I suspect that they may be missing some important neuropathological alterations.
- 12) Minor point, but the schematic in Figure 2A is not so accurate, unless this is supposed to show an early postnatal cerebellum. For instance, the climbing fibers should extend into the Purkinje cell dendrites, and the Purkinje cell dendrites that are shown have a very immature looking structure.

Reply to Reviewers comments:

General

1. General note to the revisions

- (1) We performed new experiments and analyses, and added new figure panels.
- (2) We have listed all revisions for the figures in the Overview table below.
- (3) We have highlighted by underline the major changes and additions in the revised manuscript according to our response to Reviewers' comments.

2. Overview of Revisions on Figures and displays

Figures/displays	Revision Panel	Revision Text	Content	Reviewers comments
Fig. 1	New b	-	Replace by better images	Reviewer 2 point 10
Fig. 2	New a	-	Replaced by a new scheme	Reviewer 2 point 12
Fig. 4	Complete new panel a-i	Page 8, parag. 2-3	CA3 epileptiform activity	Reviewer 2 point 4
Fig. 8	New c	Page 12, parag. 2, bottom	IP-WB on interaction of ATR and SYT2 and PROT in hippocampi and cerebella	Reviewer 2 point 7
Suppl Fig 1	New c	Page 5, parag. 1	Quantification of PCs in individual lobes of cerebella	Reviewer 2 point 11
Suppl Fig 2	New a	Page 7, parag. 1	Nissl staining of brains of 10-months old mice showing a normal morphology	Reviewer 2 points 2
Suppl Fig 3	New a	Page 7, parag. 2	Astrogliosis in the cortex area	Reviewer 2 point 2
	New d	Page 9, parag. 2	10 Hz measurements	Reviewer 2 point 2
Suppl Fig 4	c	Page 12, parag. 2	Original Fig 7C	Reviewer 2 point 7
Suppl Video	New video 3	Page 7, parag. 2	Seizure phenotype of mutant mice	Reviewer 2 points 2, 4
	New video 4	Page 7, parag. 2	Seizure phenotype of mutant mice	Reviewer 2 point 2, 4
	New video 5	Page 7, parag. 2	Seizure phenotype of mutant mice	Reviewer 2 point 2, 4

Response to Reviewers comments

[Our responses are in blue. The revised figures, text and references are quoted in red in our responses. Please note that the “page and paragraph numbers” used in this Response might be shifted due to conversion of the manuscript text file by the submission system (we had experienced that). If so, we are very sorry.]

Reply to Reviewer #1

In this paper Wang and colleagues model ATR deficiency in mice by generating cell and tissue specific deletions. This is important because hypomorphic mutations in ATR cause growth retardation, microcephaly and intellectual disability which suggests a role for ATR in non-dividing neurons, yet this role remains unknown. The authors describe a series of elegantly designed experiments to address this question, which led to a new concept - ATR deletion in post-mitotic neurons does not impact neuronal cell survival but rather causes aberrant firing leading to ataxia and epilepsy. The authors then employ proteomic and co-immunoprecipitation assays to show an interaction between ATR and synaptic proteins SYT2 and PROT. They further show that ATR deletions result in upregulation of SYT2 which is then aberrantly translocated to excitatory neurons, providing a mechanistic explanation of the in vivo observations. Overall the manuscript is well-written, the conclusions are novel and appropriately supported by data, which provides a new concept in the field that will be of broad interest to the readers of Nature communications. However, there are a few points that need to be addressed before the manuscript is accepted.

We appreciate this Reviewer’s positive view of our study. Below we provide our explanations and try to address this reviewer’s comments in a specific point-to-point reply.

1-. The interaction of ATR with SYT2 and PROT is kinase independent but appears to elicit a preferential impact on excitatory neurons. Does ATR adopt a different conformation in excitatory neurons compared to others or is there an adaptor protein that mediates downstream effects of the interaction, which is only present in excitatory neurons? What underlies the cell-specific consequences of the interaction? This might be difficult to address experimentally but the reader will benefit from further elaboration on this point as it is critical to the overall model in the manuscript.

We thank and appreciate the thoughts of the Reviewer - this is an interesting point. ATR has been shown to have many potential interactors (PMID: 17525332, Ref #24, #25). However, it is difficult to pin-point if the interaction partners would change the conformation in a cell type specific manner as it is suggested by the reviewer. Nevertheless, this is one of plausible explanations. We have followed this Reviewer’s suggestion and discussed these thoughts in the revised manuscript (see page 17, parag. 3).

2- How does absence of ATR upregulate SYT2 and cause aberrant translocation, which is again kinase independent? Is it likely that ATR sequester SYT2 thereby rendering it less available to translocate? In follow-up studies, it would be interesting to map the domain in ATR that interacts with SYT2 and use deletion mutants to confirm the model.

This point is also linked with point 1. Based on our data, ATR interacts with SYT2 and PROT. Our new experiments further proved their interaction *in vivo* (RevFig 7c). We have added this point in our discussion (see page 17, parag. 3).

3- Once replication stress is spared, ATR deficiency is not toxic to neurons nor required for brain development but is required for physiological synaptic function. The authors rule out a compensatory role for ATM – Does this imply that ATM is not required for synaptic function?

Our data rule out the compensatory role of ATM in our ATR deletion neuronal models in contrast to a previous report (Ref #74). A previous study showed that ATM is located in presynaptic compartment cultured neurons (Ref #26); however, there was no *in vivo* data reported. Nevertheless, we cannot rule out the role of ATM in the synaptic compartment to tune neurotransmission (Ref #26, Ref #75) and added this sentence in discussion (page 18, parag. 2, bottom).

4- Are there known ATR variants (or SNPs) that disrupt the interaction with SYT2 and is there evidence for association with neurological disease?

We are the first to show the interaction between ATR and SYT2 in neuronal cells. So far, there is no good humanized ATR mouse models to dissect the neuronal function of human ATR variants. Previously, two labs generated mouse models that expressed human variant (Ref #19; PMID: 19504344). While Regland et al. (PMID: 19504344) did not report the mouse phenotype, Murga et al. (Ref #19) reported that their mouse model suffered the global replication stress during development. No neuronal phenotypes were reported in these studies.

Reply to Reviewer #2

In this manuscript, the authors use a combination of mouse genetics, slice electrophysiology, and behavioral analyses to test the role of ATR in the brain. Previous studies fail to address the adult neural role of ATR because straight knock out mice are not viable. Therefore, using a conditional approach, they were able to look at GABAergic neuron function in the cerebellum and excitatory neuron function in the hippocampus. The authors show that loss of ATR in neurons causes changes in neuronal excitability, and the mice have an increased susceptibility for ataxia and epilepsy. The authors show that the deletion of ATR affects the presynaptic compartment, where it interacts with synaptotagmin 2 and PROT. The paper is generally well written and easy to follow. The images are very nice. The use of conditional genetics is a major plus. Despite the potentially broad interest of this paper, there are a number of issues that lower enthusiasm. In my comments, I outline the major problems that I hope will help the authors improve the paper.

We thank the reviewer for her/his positive evaluation of our manuscript and for helping us to improve the manuscript. To address her/his comments, we have performed new experiments and provided new figures in the revised manuscript. The electrophysiological measurement of epileptic activity is an important point, which is also requested by the Editor. We decided to generate a new cohort of ATR mutant and control mice to perform these experiments. We hope that our new experiments and figures as well as revisions have eased the concerns of this reviewer. Hopefully, the reviewer would agree that the manuscript has been greatly improved.

The authors indicate in the title that ATR regulates neuronal activity in vivo. Although I somewhat understand what the authors generally mean here, I think it would be fairer to eliminate the in vivo part. No in vivo recordings were actually conducted and this is exactly what someone reading this paper would assume from the title.

Our original intention was to indicate that this is a study based on primary neurons and brain slices from mouse models, not from other cell lines. We agreed and deleted “in vivo” from the title accordingly.

2) The authors have indicated in several instances “data not shown”. This is unnecessary as these data actually seem very important and should be included.

Following the suggestion, we included those critical “data not shown” data in in the revised manuscript (see Overview Table and RevSuppl Fig 2a, 3a, 3d, RevSuppl Videos 3, 4, 5), while those data that were repetitive or unnecessary (e.g. non-epileptic mouse brain, which showed same as controls) were maintained as “data not shown”. We hope this is OK. Of course, we can add those if the reviewer insists.

3) Related to the above, the whole story and depth of analysis is rather superficial. For example, the cerebellar electrophysiology essentially teaches us that the overall activity of Purkinje cells is affected. Given the use of slice electrophysiology, I would have expected a deeper analysis of Purkinje cell function. In this regard, it’s unclear what the authors findings mean in the larger context. There have now been many slice and in vivo cerebellar recording studies showing how changes in Purkinje cell firing rate and pattern underlie different models of ataxia, dystonia and tremor. My question is, how do the current results fit into those models?

Our original aim was to investigate the potential function of the key DNA damage response (DDR) regulator ATR beyond its essential role in proliferating cells. However, given the neurological phenotypes of human ATR-Seckel patients, we hypothesized that ATR might have an important function in postmitotic cells, which is likely masked in other model systems, in which ATR deletion is lethal for cells and organisms. Therefore, we deleted ATR in postmitotic neurons using two independent mouse models (ATR-PC Δ and ATR-FB Δ). Intriguingly, once we spare its essential DDR function in proliferating cells, we discover a previously unknown important and physiological function of ATR in neuronal activity. Therefore, the current study has placed ATR, a well-known key DDR molecule, as a new regulator in neuronal activity. These points are discussed in the manuscript (page 14, parag. 2) (see also our reply to point 8 below).

Our main finding in the ATR-PC Δ model is age-dependent ataxia. We used loose patch recordings of the acute slice model to evaluate PC intrinsic firing properties in isolation and found an increased firing rate of PCs, whereas spontaneous spiking regularity and spiking response after parallel fiber stimulation is unchanged. PCs are the main output neurons of cerebellar cortex and their high intrinsic firing rate and/or firing regularity are critical for cerebellar function (e.g. PMID: 27458803, Ref #29). As the reviewer pointed out, alterations of PC firing rate have been found in multiple pathological conditions. The PC firing rate depends on intrinsic excitability of PCs and the timing and interplay of parallel fiber and climbing fiber inputs onto PCs. Of note, the output from the PC to cerebellar nuclei is highly

regulated to generate a system that ensures the charge to the cerebellar nuclei neurons scales with the firing rate of the PCs. It is known that the impaired output of PCs could affect not only the cerebellar nuclei, but also directly neighboring PCs (Ref #53-55). In many genetic models of ataxia, the spontaneous PC firing rate is reduced. This is true for most models of spinocerebellar ataxia (SCA) with a broad spectrum of mutations and behavioral phenotypes (e.g. PMID: 33597269), in models of episodic ataxia (EA) with disturbed Ca^{2+} conductance (e.g. PMID: 16474392), and in models with deletion or defects of resurgent Na_v channels (e.g. PMID: 17928448). In a variety of unrelated disease models, e.g. in mice with calretinin deficiency (PMID: 10220453), in models of myotonic dystrophy (PMID: 28658620), in inflammation models (e.g. PMID: 32985479), or in stress models (PMID: 26445872) PC firing rate is increased. See our discussion in page 15, parag. 1.

In general, it is important to mention that the intrinsic excitability of PCs is influenced by voltage-gated sodium and potassium channels and by Ca^{2+} -activated potassium channels (SK and BK channels). Interestingly, specific alteration of ion channel function, e.g. blockade of D type Kv1 channels (Ref #61, #62) inhibits their control function of PC hyperexcitability thus leading to increased PC firing. In addition to those specific and direct mechanisms, ion channel activity in PC is regulated by a variety of modulating mechanisms, e.g. by metabotropic glutamate receptors, plasticity-induced mechanisms but also by homeostatic regulation in long-term changes of network activity (e.g. PMID: 28518055). In the revised manuscript, we now included a new paragraph discussing how our observations fit into these models of cerebellar dysfunction finally leading to ataxia (page 15, parag. 1, page 16, parag. 1).

It is noteworthy that our lab has several knockout mouse models specifically in Purkinje cells (PC Δ), in which many life essential genes (null mutation of these genes causes cell and embryonic lethality), such as TRRAP, MRE11 and NBS1 have been knocked out. While MRE11 and NBS1 form an DNA damage sensor complex MRN and can activate ATR depending on cell cycle status, none of these PC Δ models show the ataxia and electrophysiological phenotype (even after 20 months), as described here in the ATR-PC Δ model. In addition, TRRAP-PC Δ mice showed age-dependent ataxia due to a loss of PCs during neurodegeneration (see our recent publication Ref #66). All these genes are essential genes, but deletion of them in postmitotic neurons generated distinct PC phenotypes, highlighting the specificity of ATR in preventing PC defects. Moreover, while various PC specific gene knockout models display locomotor dysfunctions, some of which are associated with synaptic deficits or Purkinje cell loss, seizure phenotypes were absent in these mouse models (see a summary in Ref #65). Therefore, the cerebellar defects are not always a general trigger of seizure. Nevertheless, we discussed this aspect in the revised manuscript (page 16, parag. 1).

We fully agree that future *in vivo* experiments would be highly valuable for untangling the complex interplay of PCs *in vivo* activity and development of ataxic motor dysfunction in misbehaving animals. However, since we observed the ataxic phenotype in an age-dependent manner with a prominent phenotype in old (18 month) mice, in-depth experiments on cerebellar dysfunction are not feasible in the current revision. Instead, as also suggested in point 8, we rather focused on further investigating the epileptic phenotype with a new cohort of ATR-FB Δ mice. As these new data (RevFig 4) indicate, we indeed found increased epileptiform activity initiated in the CA3 region at young age (see also detailed reply to point 4 below).

4) Perhaps the most interesting part of the paper is that the mice develop seizures. However, unless I have missed it, I do not see any EEG recording or even videos showing the seizures. There is much work needed in order to appreciate what this phenotype looks like and what it actually means. For instance, what neurons/brain regions are involved in the seizures and what specific neuronal defects are driving the seizures? More generally, although the idea of these mice having seizures is exciting, the rationale behind this part of the paper and the actual goal of looking at seizures was underdeveloped. There is an exciting drive in the field showing that the cerebellum might be a key regions initiating seizure activity. It would have been nice to see such papers cited and discussed.

We thank this reviewer's appreciation of seizure phenotype and also for her/his' thoughtful comments. The appearance of seizures is indeed striking and it promoted us to study the role of ATR in neuron activity. In the original manuscript we provided several figure panels (original Fig 3C-D and Suppl. Figure 2F-G) to document the epilepsy phenotype. Following this reviewer's suggestion, we performed new experiments in a separate cohort of mice and provided new data:

1. We now provided videos in the revised manuscript to demonstrate the seizure semiology showing generalized tonic-clonic seizures. Seizures occur instantaneously and terminate spontaneously followed by a hypoactive post-ictal phase (Suppl Videos 3, 4, 5).
2. We documented in the original manuscript astrogliosis in the hippocampus (original Fig 3C). We also found a modest astrogliosis in the cortex of epileptic mice which are now included as a new figure (RevSuppl Fig 3a).
3. Most importantly, following the comment of this reviewer, we investigated ictogenesis in the ATR-FBA model. We decided to generate a cohort of ATR-FBA mutant and control animals to perform a series of new experiments to investigate in more detail the neuronal and circuit basis of epileptic seizure activity as the most striking phenotype in the ATR deletion model. Here, we performed electrophysiology recording in the CA3 region of the hippocampus. In particular, we induced seizure-like activity showing rhythmic epileptiform discharges and population bursts in a submerged chamber of acute slices using Mg²⁺-free solution containing 4 mmol K⁺ (PMID: 30525115, PMID: 3031235). Here, we found potent epileptiform activity resembling preictal epileptiform discharges (PEDs) consisting of population field activity (PFA, depicted by applying 1-100 Hz bandpass filter), fast ripples (FRA, 200-500 Hz bandpass), and multiple-unit activity (MUA, 500 Hz high pass) greatly enhanced in ATR-FBA mice. Moreover, the total power of local field potentials and, in particular, the power spectrum of isolated epileptiform discharges was increased in ATR-FBA mice. Importantly, the spike rate within those population bursts, directly reflecting CA3 neuron spiking frequency, was also enhanced in ATR-FBA. Both findings, epileptiform population bursts and fast ripples, are characteristics in areas generating spontaneous seizures (Ref #35). Since epileptiform activity often initiates in the CA3 region of hippocampus, these prominent changes suggest CA3 as the pacemaker of ictal activity in ATR-FBA. Together, these new data obtained in a standardized model of the hippocampal circuit indicate greatly enhanced susceptibility for epileptic activity already in young stages of the ATR-FBA mouse model eventually resulting in spontaneous occurring seizures. We have added these new data (RevFig 4; page 8, parag. 2-3) and discussion (page 16, parag. 2) in the revised manuscript.

While the cerebellum defect can be an origin or driver of seizures (see recent reviews Ref #63, #64), our current study and also other PC deletion mouse models do not show this link because:

- (1) our ATR-PC Δ mice do not show any seizure phenotype in the observation period of more than 20 months. As mentioned above (see reply to point 3), we have several PC Δ mouse models in the lab and, regardless of their cerebellar phenotype, we did not observe any seizure phenotype in these models in the period of even more than 20 months of age.
- (2) various PC specific gene knockout models display locomotor dysfunctions, some of which are associated with synaptic deficits or Purkinje cell loss, but lack seizure phenotypes.
- (3) ATR-FB Δ mice are ATR specific deletion in forebrain, which showed epilepsy. These mice do not have ATR deletion in the cerebellum.

In sum, together with our “reply to points 3 and 8” and revised discussion (page 15, parag. 1; page 16, parag. 1), our new experiments demonstrate (RevFig 4) that ATR has a direct function in regulation of neuronal activity and loss of ATR renders neurons vulnerable to epileptic activity.

5) I am not sure what the authors are referring to by “motor strength”.

We apologize for the wrong wording. We meant “grab strength and motor coordination/function”. We have rephrased this description (page 5, parag. 2)

6) The authors state “...to avoid the secondary effects of epileptic seizures...”. This is confusing since seizures were a part of the phenotype. But here the authors wish to avoid them. There is no rationale provided for this.

We have corrected the sentence (page 9, parag. 2).

7) Neuroblastoma cells were used for the protein-protein interaction studies. What happens *in vivo*? And specifically, what do these interactions look like in the cerebellum versus the hippocampus?

It is an important point. Following this suggestion, we have performed IP experiments using cerebellar and hippocamp tissues. Because ATR antibodies (all tested) are problematic in IP (in our hands and also known in the field), we used SYT2 and PROT antibodies to pull down ATR in both cerebellar and hippocampal tissues and confirmed the interaction of ATR with SYT2 and PROT *in vivo*. These new data are included in RevFig 8c. In our original manuscript (original Fig 7C) we showed ATR interaction with SYT1 in the hippocampal tissue, which is now moved to RevSuppl Fig S4c.

8) Overall, the reason for a split of the electrophysiology into cerebellar and hippocampal sections is unclear. Each system is given a somewhat superficial treatment and as a result I am really unsure what we have learned. Why not dig into one of these systems a little deeper? Again, please see my above comment about other models of cerebellar function in ataxia. We know so much about this topic and littler of the prior knowledge is exploited here to advance the understanding of ataxia or cerebellar function.

As described in response to point 3, our study came from the original motivation to understand the non-canonical function of ATR, which is beyond its DDR role in proliferating

cells. Therefore, we engineered two mouse models to delete ATR in postmitotic neurons, namely inhibitory (PCs) and excitatory neurons (CaMKII positive neurons), which can spare its essential DDR function in proliferating cells. Interestingly and intriguingly, although these two models show different neuronal phenotypes, ATR has a general impact on the synaptic compartment and neuronal activity. The important point is that although ATR has been acclaimed to be essential DDR molecule, our study demonstrates an important and physiological function of ATR in postmitotic neurons.

We are aware of the limitation of the models that were used. The problem for the molecular pathway analysis in Purkinje cell specific knockout is the relative scarcity of Purkinje cells (PC) with respect to the overall cell number in the cerebellum. Therefore, we switched to a forebrain-deleted model to analyze the pathway controlled by ATR for the feasibility. To gain insight we performed proteomics and found specific molecular targets affected by ATR, i.e., SYT2 and PROT. We provided detailed analyses on these proteins in brain tissues as well as in neural cells (N2A) for its molecular mechanism. We further confirmed ATR interaction with these two molecules in both hippocampus and cerebellar tissues (RevFig 8c). However, we have no direct evidence on the same mechanism governed by ATR in PC presynaptic activities or firing rate. At the moment, we do not have good hypothesis to reconcile how ATR plays a general role in neuronal excitability. However, the fact that ATR deletion has been shown to soften the nuclear and cytoplasmic membrane (PMID: 25083873, PMID: 32973141), encourages us to imagine that ATR's involvement in membrane-bound channel complexes might contribute to neuronal excitability. Naturally, this point is too speculative and we are not comfortable to explicitly discuss it in the current manuscript. Nevertheless, this hypothesis constitutes an interesting project in future.

Following reviewer's suggested, we performed a completely new series of experiments and investigated the neuronal and circuit basis of epileptic seizure activity in more detail (see also detailed responses to question 4). Here, we found that seizure-like activity initiated in the CA3 region at young age shows increased power of preictal epileptiform discharges including population field activity (PFA), fast ripple activity (FRA) as well as increased frequency of multiple unit activity (MUA) (see new Fig 4, results (page 8, parag. 2-3), and discussion (page 16, parag. 2)).

With regard to cerebellar dysfunction in the ATR-PC Δ model we have added a new paragraph in the revised manuscript to integrate our findings of an increased PC firing rate into the context of models of ataxia (page 15, parag. 1; page 16, parag. 1, see also more detailed explanations in our reply to point 3).

9) The Discussion section is lacking many essential arguments. For instance, there is no discussion about the seizure phenotype, no discussion about what the cerebellar dysfunction means or any potential mechanisms therein, and there is no discussion about how the synaptic and molecular defects potentially drive cerebellar-dependent motor behaviors.

As also indicated in our responses to points 3, 4 and 8, we have added the following new paragraphs in the discussion:

1. Discussion of the seizure phenotype, semiology and potential mechanisms (page 16, parag. 2).
2. Discussion of the cerebellar phenotype and potential mechanisms (page 15, parag. 1).

10) In Figure 1B, it's not so convincing to me that there are no cerebellar defects. Subtle as it might be, I see defective Bergmann glia processes. Also, for both the control and the mutant, the calbindin staining could be better quality.

We did not find obvious morphological defects as described in the text. The original images in Fig 1B might be not high quality due to conversion of figures after loading figures to the submission system. Apologies. To improve it, we now provide new and better figures to replace the original Fig 1B (see RevFig 1b).

11) In Figure 1, its peculiar to me that the authors chose to examine lobule X. This lobule is well known to have a neuroprotective nature in many different diseases (see the older work from Richard Hawkes, but you can also see the newer work from Zoghbi, Shakkottai, and other labs). Thus, perhaps the authors need to look at other lobules as I suspect that they may be missing some important neuropathological alterations.

This points is perhaps stemmed from our insufficient description. We actually examined all lobules of the cerebellum and found no obvious difference. (1) The lobule X was just to show a representative region for PCs. As a matter of fact, the original Fig 1C showed a quantification of all PCs from all lobules of cerebella. To easy the concerns, now we provide the quantifications of individual lobules in RevSuppl Fig 1c. As shown, we do not see obvious difference between ATR knockout and controls cerebella throughout the lifespan from 3 to 20 months.

12) Minor point, but the schematic in Figure 2A is not so accurate, unless this is supposed to show an early postnatal cerebellum. For instance, the climbing fibers should extend into the Purkinje cell dendrites, and the Purkinje cell dendrites that are shown have a very immature looking structure.

Thank you; we have modified the figure.

Reviewers' Comments:

Reviewer #1:

Remarks to the Author:

The authors provided convincing justifications which adequately addressed my comments. I also note the additional experimental data, which have strengthened the overall conclusions of this study.

Reviewer #2:

Remarks to the Author:

The authors have done an outstanding job in addressing the comments raised upon the initial submission. This reviewer appreciates the thorough and clear responses to every comment. The additional data that have been provided clear up many issues that were brought up. This is a very interesting piece of work.